# A homozygous loss-of-function mutation leading to CYBC1 deficiency causes chronic granulomatous disease

Gudny A. Arnadottir [1], Gudmundur L. Norddahl[1], Steinunn Gudmundsdottir[1], Arna B. Agustsdottir[1], Snaevar Sigurdsson[1], Brynjar O. Jensson[1], Kristbjorg Bjarnadottir[1], Fannar Theodors[1], Stefania Benonisdottir[1], Erna V. Ivarsdottir [1,2], Asmundur Oddsson [1], Ragnar P. Kristjansson[1], Gerald Sulem[1], Kristjan F. Alexandersson[1], Thorhildur Juliusdottir[1], Kjartan R. Gudmundsson[1], Jona Saemundsdottir[1], Adalbjorg Jonasdottir[1], Aslaug Jonasdottir[1], Asgeir Sigurdsson[1], Paolo Manzanillo[1], Sigurjon A. Gudjonsson[1], Gudmundur A. Thorisson [1], Olafur Th. Magnusson[1], Gisli Masson[1], Kjartan B. Orvar[3,4], Hilma Holm[1], Sigurdur Bjornsson[3,4], Reynir Arngrimsson[5,6], Daniel F. Gudbjartsson [1,2], Unnur Thorsteinsdottir[1,6], Ingileif Jonsdottir[1,6], Asgeir Haraldsson[6,7], Patrick Sulem [1] & Kari Stefansson[1,6]

Mutations in genes encoding subunits of the phagocyte NADPH oxidase complex are recognized to cause chronic granulomatous disease (CGD), a severe primary immunodeficiency. Here we describe how deficiency of CYBC1, a previously uncharacterized protein in humans (C17orf62), leads to reduced expression of NADPH oxidase's main subunit (gp91$^{phox}$) and results in CGD. Analyzing two brothers diagnosed with CGD we identify a homozygous loss-of-function mutation, p.Tyr2Ter, in *CYBC1*. Imputation of p.Tyr2Ter into 155K chip-genotyped Icelanders reveals six additional homozygotes, all with signs of CGD, manifesting as colitis, rare infections, or a severely impaired PMA-induced neutrophil oxidative burst. Homozygosity for p.Tyr2Ter consequently associates with inflammatory bowel disease (IBD) in Iceland ($P = 8.3 \times 10^{-8}$; OR = 67.6), as well as reduced height ($P = 3.3 \times 10^{-4}$; −8.5 cm). Overall, we find that CYBC1 deficiency results in CGD characterized by colitis and a distinct profile of infections indicative of macrophage dysfunction.

[1] deCODE Genetics/Amgen, Inc., Reykjavik, Iceland. [2] School of Engineering and Natural Sciences, University of Iceland, Reykjavik, Iceland. [3] Department of Internal Medicine, Landspitali University Hospital, Reykjavik, Iceland. [4] The Medical Center, Glaesibae, Reykjavik, Iceland. [5] Department of Genetics and Molecular Medicine, Landspitali University Hospital, Reykjavik, Iceland. [6] Faculty of Medicine, University of Iceland, Reykjavik, Iceland. [7] Children's Hospital Iceland, Landspitali University Hospital, Reykjavik, Iceland. These authors contributed equally: Gudny A. Arnadottir, Gudmundur L. Norddahl. Correspondence and requests for materials should be addressed to P.S. (email: patrick.sulem@decode.is) or to K.S. (email: kari.stefansson@decode.is)

Chronic granulomatous disease (CGD) is a rare primary immunodeficiency characterized by severe recurrent bacterial and fungal infections, along with manifestations of chronic granulomatous inflammation[1]. The incidence of the disease varies significantly worldwide, from around 1 in 200,000 in North America and Europe, to 1 in 70,000 in the Israeli Arab population[2]. CGD results from an impaired ability of phagocytes to mount a burst of reactive oxygen species (ROS) in response to pathogens[2]. The production of ROS is catalyzed by a multi-protein enzymatic complex, known as the NADPH oxidase complex[3]. A majority of CGD patients carry pathogenic genotypes in one of five genes encoding subunits of the phagocyte NADPH oxidase; either X-linked recessive mutations in *CYBB*, or autosomal recessive mutations in *CYBA*, *NCF1*, *NCF2*, or *NCF4*[2,4–6]. Patients with X-linked recessive mutations in *CYBB* are generally recognized to have the most severe disease course with earlier age at onset, whereas patients with autosomal recessive mutations in *NCF1* show a significantly higher age at onset[1,2,7]. In a recent review of *NCF4* mutations, the clinical presentation of NCF4-deficient patients is described as being even more distinct, resembling a mild, atypical form of CGD[6].

For the purpose of searching for associations between variants in the sequence and phenotypes, we have whole-genome sequenced (WGS) 37K Icelanders and genotyped 155K, a large fraction of the Icelandic population (11 and 46% of 338K, respectively)[8–10]. This dataset allows for accurate detection of genotypes down to a frequency of 0.01% in all 155K[8,9]. Moreover, by serving as a population reference, the set is proving instrumental for genetic analysis of rare diseases in the clinical setting[8,11,12].

Through WGS of two brothers diagnosed with CGD, we identify a homozygous loss-of-function mutation, p.Tyr2Ter, in *CYBC1* (previously *C17orf62*). In our existing set of sequenced and chip-genotyped Icelanders we find six additional homozygous individuals. Genotype-based recall confirms that all eight homozygotes have signs of CGD, manifesting as colitis, rare infections, or a severely impaired PMA-induced neutrophil oxidative burst. We show that homozygosity for p.Tyr2Ter results in complete loss of CYBC1, and reduced expression of gp91$^{phox}$, NADPH oxidase's main subunit, leading to an impaired oxidative burst. Our results indicate that CYBC1 is essential for successful formation of the NADPH oxidase complex, likely by acting as a chaperone. An excess of colitis in the group of homozygotes, as well as a distinct profile of infections, suggests that CYBC1 deficiency has a more pronounced effect on macrophages. In summary, by leveraging an extensive population database we elucidate the role of a previously uncharacterized gene in humans and identify a novel cause of CGD.

## Results

### Analysis of two brothers diagnosed with CGD.
We were presented with two Icelandic brothers diagnosed with CGD (individuals A and B, Fig. 1a and Table 1), to search for the causal mutation through WGS. The brothers were initially diagnosed with Crohn's disease (CD), at 7 and 9 years of age, after developing severe diarrhea with intestinal biopsies showing granulomatous lesions compatible with CD. The younger brother had developed episodes of acute lymphadenitis and inflammation of the orbit at the ages of 5 and 6 years, respectively. Both brothers experienced recurrent bacterial infections, including infections around the nose and an active wound infection in the mouth, from which the opportunistic bacterium *Burkholderia cepacia* was cultured. *B.cepacia* is known to cause infections in immuno-compromised hosts, in particular CGD patients[4]. At this stage, the combined clinical, histological, and bacteriological evidence

led to a suspicion of CGD. A formal CGD diagnosis was subsequently confirmed for both brothers based on PMA-induced neutrophil oxidative burst tests at the time (Fig. 1b). The number of infections reported for the brothers was nonetheless somewhat less than what would be expected in X-linked CGD patients. The brothers' gastrointestinal symptoms did not respond to conventional treatment for Crohn's disease, and were the main reason for them undergoing hematopoietic stem cell transplantation (HSCT) at 17 (individual A) and 18 (individual B) years of age. The older brother (individual B) died of post-HSCT complications; whereas, the younger brother (individual A) was successfully transplanted in 2010 and has been symptom-free since then (see Supplementary Note 1 for full clinical description).

We sequenced the whole genomes of the two brothers (DNA samples taken pre-HSCT), hereafter referred to as the probands, their three unaffected siblings and parents (see pedigree Fig. 1a, and Methods section). We found no rare coding or splice-site mutations in the five genes known to harbor mutations causing CGD (all five genes were well-covered in the probands' sequence data, Supplementary Table 1). Previously, we had identified a known mutation in *CYBA*[13] in Iceland (NP_000092.2:p. Arg90Trp, MAF = 0.58%), under a recessive mode of inheritance. We found five Icelanders homozygous for that mutation (out of 155K chip-genotyped), two were diagnosed with CGD, and three had a clinical profile consistent with CGD (Supplementary Table 2). Importantly, WGS followed by Sanger sequencing verified the absence of this *CYBA* mutation from the two probands. We subsequently expanded our analysis to the coding and splicing regions of all RefSeq genes (*n* = 18,570). Due to the rarity of the disease and high penetrance of known CGD mutations we focused on rare genotypes, defined using our reference set of the Icelandic population (37,260 individuals WGS to a median depth of 38×). We selected heterozygous genotypes with a minor allele frequency (MAF) below 0.05%, excluding genotypes carried by either of the two unaffected parents, and homozygous/compound heterozygous genotypes where variants had a MAF below 2%, including both X-linked and autosomal genotypes. We found two genotypes fulfilling these criteria, in genes not previously known to cause Mendelian disease. One of these, a homozygous missense mutation in *GCGR* encoding the human glucagon receptor (NM_000160.3:c.449G>A; NP_000151.1: p.Ser150Asn; hg38 position chr17:81,811,277; MAF = 0.33%), was dismissed based on lack of biological relevance (Supplementary Note 2). The other corresponds to a predicted complete knockout, a homozygous stop-gained mutation, p.Tyr2Ter, in *CYBC1* (NM_001033046.3:c.6C>G; NP_001028218.1:p.Tyr2Ter; hg38 position chr17:82,449,249; MAF = 0.76%). Prior to our application for the gene symbol *CYBC1*, it was known under the placeholder symbol *C17orf62*. The brothers are the only members of their family homozygous for *CYBC1* p.Tyr2Ter, their parents and two unaffected siblings are heterozygous carriers, and their third sibling a non-carrier (all confirmed by Sanger sequencing). CYBC1 shares 89% amino acid sequence identity with the murine bc017643 which was recently shown to be essential for ROS production[14], consistent with the reduced ROS production described for CGD patients. Both neutrophils and BM-derived macrophages from *bc017643*-knockout mice showed a highly impaired oxidative burst in response to a range of stimuli, including PMA. Accordingly, *bc017643*-knockout mice (-/-) showed high susceptibility to bacterial infections, evidenced by *Listeria monocytogenes* and *Salmonella enterica* serovar Typhimurium infections[14]. Interestingly, while macrophages from the *bc017643*-knockout mice had a similar ROS deficit as macrophages from gp91$^{phox}$/Cybb-deficient mice, neutrophils from the *bc017643*-knockout mice showed some ROS generation whereas neutrophils from the gp91$^{phox}$/Cybb-deficient mice did not[14].

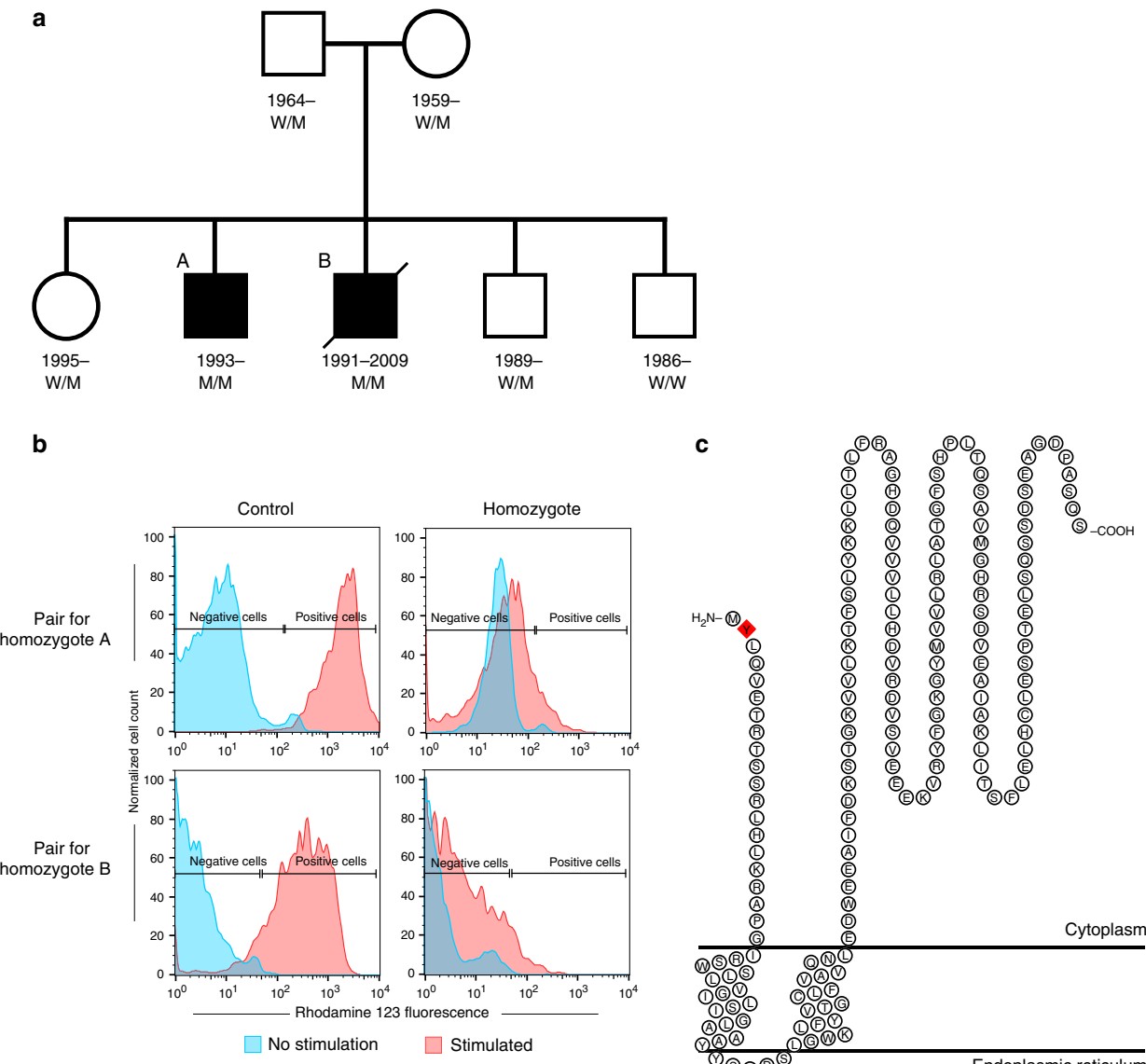

**Fig. 1** Pedigree and burst test results for the two probands, and the CYBC1 protein. **a** Pedigree of the two CGD brothers showing their *CYBC1* p.Tyr2Ter genotypes. Squares represent males, circles represent females, and a slashed symbol indicates a deceased individual. Filled symbols represent affected individuals; the two affected are referred to as individuals A and B in the manuscript. The genotype of the p.Tyr2Ter mutation (NP_001028218.1:p.Tyr2Ter; NM_001033046.3:c.6C>G; hg38 position chr17:82,449,249) is indicated with M and W, M representing the mutated allele and W the wild type. M/M therefore indicates homozygous status, W/M indicates heterozygous status and W/W a non-carrier. **b** Neutrophil oxidative burst test for the two CGD brothers homozygous for *CYBC1* p.Tyr2Ter (individuals A and B) and their controls, test was performed pre-HSCT. Left panel shows fluorescent peaks for unstimulated and PMA stimulated neutrophils from the controls, and the right panel shows peaks for unstimulated and PMA stimulated neutrophils from the two homozygous brothers. Negative and positive cells are defined by setting a gate for unstimulated cells. Neutrophils from individuals A and B failed to generate an oxidative burst equivalent to their controls, their respective stimulation indices were $SI_A = 1.34$ and $SI_B = 2.50$. **c** Topological prediction of CYBC1 (NP_001028218.1)[28]. CYBC1 is predicted to be a transmembrane protein, spanning the lipid bilayer via two transmembrane regions (aa 21–39 and aa 45–63). A red diamond represents the p.Tyr2Ter mutation at the second amino acid of the protein

The availability of population level data allowed us to search for other individuals homozygous for *CYBC1* p.Tyr2Ter. Out of a learning set of 28,075 WGS Icelanders we detected two other homozygous individuals, as well as 372 heterozygous carriers of the mutation. This allowed for accurate imputation (Methods section) of the mutation into our set of 155K chip-genotyped Icelanders based on long-range haplotype sharing[8] (imputation information = 0.99). Through imputation we found four additional individuals homozygous for p.Tyr2Ter, and confirmed their genotypes with Sanger sequencing.

**Colitis and rare infections in p.Tyr2Ter homozygotes**. To investigate the consequences of *CYBC1* p.Tyr2Ter, we searched for pathological manifestations in these six additional homozygotes and found that three had colitis, like the two probands, and had received a diagnosis of inflammatory bowel disease (IBD) (Table 1). Intestinal inflammation is a common manifestation of CGD and around 40% of CGD patients develop CD-like problems[15]. Of the three additional homozygotes suffering from colitis, two had a confirmed granulomatous colitis (individuals C and F). Individual C was initially diagnosed with ulcerative colitis

**Table 1 Phenotypes of eight *CYBC1* p.Tyr2Ter homozygous individuals**

| ID | Sex | YOB/D | GI symptoms | Burst test[a] | Height[b] | Infections | Immunological |
|---|---|---|---|---|---|---|---|
| A[c] | M | 1993 | CD (7); Mouth ulcer, *B.cepacia* Anal abscess, anal fistula, anal fissure (15); Infectious diarrhea (16) | Abnormal[a] | 172 (18) | Legionnaires' disease, *Legionella* (14); Local infections of skin, subcutaneous tissue (repeated); Viral infection (12); Accute suppurative otitis media (22) | Acute inflammation of orbit (6); Acute lymphadenitis of face, head and neck (5); Acute pancreatitis (16); Pneumonia (16) |
| B[c] | M | 1991/ 2009 | CD[d] (9); Cutaneous abscess, furuncle and carbuncle of buttock (16) | Abnormal[a] | 162 (17) | Bacterial intestinal infection, *C. difficile* (16); Candidal septicaemia, *C.albicans* (18); Viral intestinal infection (16); Suppurative otitis media (6) | Hepatosplenomegaly (17); Acute pancreatitis (18) |
| C | M | 1985 | CD (12); Anal fissures (4, 7); Anal abscess (11) | ND | 180 (25) | — | — |
| D | F | 1985 | — | Abnormal[a] | 159 (30) | Herpes[e] (25) | RF positive (21); Eczema[e] |
| E | M | 1980 | UC (14) | ND | 178 (37) | Onychomycosis (repeated) | — |
| F | M | 1980 | CD[d] (19) | ND | 173 (37) | Infectious diarrhea (34); Acne[e] (chronic); Herpes[e] (15); Shingles, varicella zoster[e] | Chronic tonsillitis (5); Adenoid vegetations (5); Chronic nephritic syndrome (26) |
| G | F | 1955/ 2015 | — | ND | 159 (50) | Invasive pneumococcal disease, *S. pneumoniae* (positive blood culture) (30) | Interstitial pulmonary disease with fibrosis (56) |
| H | F | 1940 | — | ND | 156 (55) | Miliary tuberculosis, *M.tuberculosis* (13) | — |

Figures in parentheses () denote age at diagnosis or measurement, in years
*YOB/D* year of birth/death (rounded by 5 years for individuals C–H), *GI* gastrointestinal, *CD* Crohn's disease, *UC* ulcerative colitis, *CGD* chronic granulomatous disease, *ND* not determined, *RF* rheumatoid factor
[a]Burst test for individual A and B was performed in 2008, burst test for individual D was performed in 2017
[b]All heights are given in cm. Average and SD for height of Icelandic males and females is 178.8 ± 6.9 cm and 165.6 ± 6.3 cm, respectively[35]
[c]Individuals A and B are brothers, presented in Fig. 1
[d]Individuals B and F underwent total colectomy
[e]Self-reported phenotypes (via an online questionnaire)

(UC), receiving a CD diagnosis only after undergoing total colectomy that revealed extensive granulomatous inflammation of the colon. The third individual, individual E, was diagnosed at 14 years of age with total colitis. Repeated biopsies from colon did not yield granulomatous changes. However, even though granulomas were not observed in individual E, his clinical course could be considered to better fit CD than UC. Notably, granulomatous changes are unconfirmed in the majority of CGD patients with colitis[16]. Furthermore, diffuse colitis involving all regions of the colon, as seen for individual E, could be considered characteristic of the gastrointestinal involvement in CGD[16]. As expected, p. Tyr2Ter associates with IBD in Iceland under the recessive model ($P = 8.3 \times 10^{-8}$; OR (95% CI) = 67.6 (14.5, 315.5); $n_{cases} = 2429$; $n_{controls} = 338,647$; likelihood ratio test; Supplementary Table 3). Limiting to chip-typed individuals, we found that heterozygous carriers of p.Tyr2Ter were not at a significantly higher risk of IBD than non-carriers ($P = 0.31$; OR (95% CI) = 1.21 (0.84, 1.75); $n_{cases} = 1701$; $n_{controls} = 135,106$; likelihood ratio test).

To gather additional clinical and biological information we attempted genotype-based recall of p.Tyr2Ter homozygotes for phenotyping. We obtained additional hospital-based information (including discharge diagnoses and laboratory measurements, see Table 1 and Methods section), and found that seven out of eight homozygotes had experienced severe and/or early-onset bacterial or fungal infections (including the two probands). Individual D was the only one not to have reported any severe infections, at 30 years of age. In patients with CGD, microbial catalase is traditionally recognized as the most important virulence factor in infections[17]. Consistent with this, many of the infections noted in the homozygous individuals were caused by catalase-positive pathogens, including a mucous membrane infection by *Burkholderia cepacia*[2], candidal septicemia (*Candida albicans*[18]), Legionnaires' disease (*Legionella*[19]), and miliary tuberculosis (*Mycobacterium tuberculosis*[20]). The rarity of most of these infections is notable, only 61 occurrences of *Legionella* infections have been reported in Iceland between 1997 and 2017, and only 19 individuals have been diagnosed with candidal septicemia over the same period[21]. Thus, it is noteworthy that individual A developed both Legionnaires' disease and candidal septicemia. Similarly, only 55 occurrences of miliary tuberculosis have been noted in Iceland, including individual H (out of 11,438 tuberculosis cases, diagnosed between 1901 and 1989[22]). Individual G had invasive pneumococcal disease at 30 years of age caused by *Streptococcus pneumoniae*[7], and later developed interstitial pulmonary fibrosis, a noted complication of the exaggerated inflammatory response in CGD[2,23,24].

In terms of microbe spectrum, the overall infectious profile of the homozygous individuals could be considered somewhat unusual for CGD. The invasive fungal sepsis by *C.albicans*, and the miliary tuberculosis by *M.tuberculosis* are characteristic of CGD, as well as the pneumonia episodes reported for individual A (at 14 years of age, registered as Legionnaires' disease, and again at 16 years of age). However, the pathogens cultured from the pneumonia episodes are only sporadically reported in CGD (*Legionella*[1,25] and *S.pneumoniae*[7]), and the absence of some of the most frequently encountered pathogens in CGD, such as *Staphylococcus aureus* and *Aspergillus* species, is noteworthy[2,7].

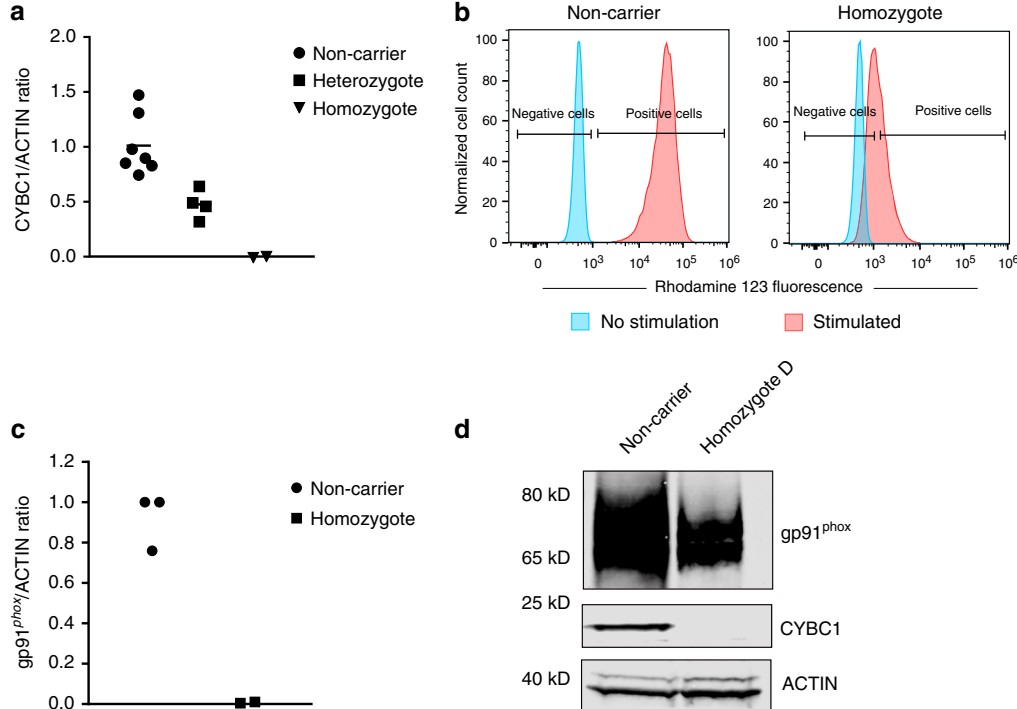

**Fig. 2** Effect of p.Tyr2Ter on CYBC1, the neutrophil oxidative burst, and gp91[phox]. **a** CYBC1 protein expression, relative to ACTIN, in lymphocytes from *CYBC1* p.Tyr2Ter homozygous individuals D and F (triangles), four heterozygous carriers (squares), and matched (age and sex) non-carriers (circles). CYBC1 expression was not detected in the homozygous individuals in contrast to their matched non-carriers, and was reduced by 53% in the heterozygous carriers, compared to matched non-carriers. The analysis was performed by western blot (shown in Supplementary Fig. 3 and 4), where ACTIN was used as a loading control. **b** Neutrophil oxidative burst test for *CYBC1* p.Tyr2Ter homozygous individual D and her matched (age and sex) non-carrier. Left panel shows fluorescent peaks for unstimulated and PMA stimulated neutrophils from the non-carrier, and the right panel shows peaks for unstimulated and PMA stimulated neutrophils from homozygous individual D. Negative and positive cells are defined by setting a gate for unstimulated cells. Neutrophils from individual D failed to generate an oxidative burst equivalent to her matched non-carrier, the stimulation index for homozygous individual D was $SI_D =$ 2.35. **c** Protein expression of gp91[phox], relative to ACTIN, in monocyte-derived macrophages from *CYBC1* p.Tyr2Ter homozygotes D and F (squares) and their matched (age and sex) non-carriers (circles). gp91[phox] expression was absent in the two homozygotes in contrast to their matched non-carriers. The analysis was performed by western blot (shown in Supplementary Fig. 5), where ACTIN was used as a loading control. **d** CYBC1 and gp91[phox] expression in fresh neutrophils from *CYBC1* p.Tyr2Ter homozygous individual D and her matched (age and sex) non-carrier. CYBC1 expression was not detected in the homozygous individual in contrast to the non-carrier. gp91[phox] expression was ~50% lower in the homozygote than her matched non-carrier. The analysis was performed by western blot, and ACTIN was used as a loading control

In addition to the two probands, we performed WGS for the six other *CYBC1* p.Tyr2Ter homozygotes. The five individuals with gastrointestinal involvement were analyzed for variants previously known to cause monogenic IBD[15,26], and the absence of such variants confirmed. Additionally, no known or expected pathogenic variants were detected in other known CGD genes in any of the eight p.Tyr2Ter homozygotes. Consistent with the effect of other known CGD genotypes on stature[4] we also found that homozygosity for p.Tyr2Ter associates with reduced height ($P = 3.3 \times 10^{-4}$; effect (95% CI) = −1.24 SD (−1.92, −0.56)/ −8.5 cm; likelihood ratio test; Supplementary Tables 3 and 4). Taking all of the above into consideration, we conclude that the clinical presentation of seven out of eight homozygous individuals (all but D) is suggestive of CGD.

**Functional consequences of p.Tyr2Ter.** *CYBC1* encodes a 187 amino acid protein (NP_001028218.1, canonical as defined by RefSeq), highly expressed in whole blood[14,27]. It is predicted to be a transmembrane protein, spanning the lipid bilayer via two transmembrane regions (Fig. 1c)[28]. The p.Tyr2Ter mutation introduces a premature termination codon at the second amino acid of the protein, preventing translation of the full-length

protein. We observed no reduction in *CYBC1* mRNA expression in white blood cells from heterozygous carriers of p.Tyr2Ter ($n_{carriers} = 29$; $n_{non-carriers} = 2430$; $P = 0.86$; Effect (95% CI) = 0.03 SD (−0.36, 0.30), Supplementary Fig. 1), nor did we observe any signs of allelic imbalance when assessing allele-specific expression (Supplementary Fig. 2). This indicates that p.Tyr2Ter does not lead to nonsense-mediated mRNA decay (NMD), consistent with NMD being less efficient when premature termination codons occur in such close proximity to the start codon[29]. Rather, we expect p.Tyr2Ter to result in a lack of product at the protein level. We investigated the possibility of an alternative start codon in *CYBC1* producing a functional protein. The closest codon for methionine following the same reading frame as p.Tyr2Ter occurs in exon 6 (out of 7 exons). The non-coding mRNA corresponding to this start codon (NR_036518, expressed in whole blood[27]) encodes a sequence of 63 amino acids, lacking the two membrane-spanning domains of CYBC1.

To study the direct effect of p.Tyr2Ter at the protein level, we performed western blot analysis on available lymphocytes from four heterozygous carriers of p.Tyr2Ter, revealing 53% less CYBC1 protein than in matched non-carriers (Fig. 2a). Additionally, through genotype-based recall we obtained blood

samples from two p.Tyr2Ter homozygotes (individuals D and F, Table 1). We found no trace of the full-length CYBC1 protein in lymphocytes from the two homozygotes (Fig. 2a).

**Chaperone for cytochrome b-245.** The oxidative burst consists of a rapid production and release of ROS, a process catalyzed by the NADPH oxidase complex[3]. The phagocyte NADPH oxidase is composed of six subunits, three cytosolic components (p47$^{phox}$, p67$^{phox}$, and p40$^{phox}$), a regulatory G-protein (Rac1/Rac2) and an essential membrane-bound heterodimer (gp91$^{phox}$-p22$^{phox}$), known as cytochrome b-245, which mediates the transfer of electrons to oxygen[3]. The dimerization of gp91$^{phox}$ and p22$^{phox}$ is a relatively inefficient process, and as unassembled monomers the two proteins quickly undergo degradation[14,30,31]. Neutrophils and bone marrow (BM)-derived macrophages from *bc017643*-knockout mice had markedly reduced expression of both gp91$^{phox}$ and p22$^{phox}$, despite expressing normal mRNA levels for these proteins, most likely due to the role of bc017643 as a chaperone in the dimerization of cytochrome b-245. The cytosolic components of NADPH oxidase (p47$^{phox}$, p67$^{phox}$, and p40$^{phox}$) were not differentially expressed in *bc017643*-deficient mice[14]. To determine whether a similar role could be expected for CYBC1 in humans, we measured gp91$^{phox}$ in monocyte-derived macrophages and neutrophils from p.Tyr2Ter homozygotes. We detected no gp91$^{phox}$ in monocyte-derived macrophages from homozygous individuals D and F (Fig. 2c), in contrast to matched non-carriers, and a ~50% reduction in gp91$^{phox}$ levels in neutrophils from individual D (neutrophils were not available from individual F, Fig. 2d). Furthermore, we were able to test and confirm an impaired PMA-induced oxidative burst in neutrophils from homozygous individual D (Fig. 2b), thereby providing independent confirmation of an impaired oxidative burst in a p.Tyr2Ter homozygote (as observed previously for the two probands, Fig. 1b). Individual D is therefore positive for the test widely used as a diagnostic test for CGD without having, to our knowledge, developed colitis or chronic infections at age 30 (Table 1 and Methods section). Our findings, together with the prior observation in mice, suggest a chaperone role for CYBC1 in the dimerization of gp91$^{phox}$ and p22$^{phox}$ in humans. We therefore renamed the gene *CYBC1*, for cytochrome b-245 chaperone 1.

The p.Tyr2Ter mutation is absent from 138,632 exomes/genomes in a large multi-ethnic public database (gnomAD), and no other coding homozygous loss-of-function mutation is listed in the canonical transcript on gnomAD[32] (Supplementary Table 5). Nonetheless, we came across a homozygous missense mutation in *CYBC1*, p.Asp43Asn (NM_001033046.3:c.127G>A, hg38 position chr17:82,447,580), in a single case in a recent publication from Saudi Arabia[33]. Although the mutation was classified as a missense, we identified its location to be at the exonic splice junction where it is predicted to disrupt correct splicing of the mRNA[34]. Interestingly, the individual homozygous for that mutation had symptoms compatible with CGD, including lymphadenopathy, hepatosplenomegaly, recurrent fever, and growth retardation[33].

**Discussion**
In summary, among 155K chip-genotyped Icelanders we identified eight homozygous for a rare loss-of-function mutation, p.Tyr2Ter, in *CYBC1* (previously known as the uncharacterized *C17orf62*). The eight homozygotes show signs compatible with CGD, either gastrointestinal symptoms, rare infections, and/or a biochemically proven oxidative burst defect.

Impaired expression of gp91$^{phox}$ in p.Tyr2Ter homozygous individuals, the oxidative burst defect, and prior evidence in knockout mice all point to CYBC1 as an essential factor for successful formation of the NADPH oxidase complex. CYBC1 is known to co-localize and interact with gp91$^{phox}$ in the endoplasmic reticulum (ER)[14,35], where it likely acts as a chaperone for dimerization of gp91$^{phox}$ and p22$^{phox}$ (cytochrome b-245). Although no definitive conclusion can be drawn from only one individual, the difference in expression of gp91$^{phox}$ between monocyte-derived macrophages (absent) and neutrophils (50% reduced) from homozygous individual D is noteworthy, and suggests the presence of more than one chaperone for cytochrome b-245 in the neutrophil lineage. Consequently, mechanisms more specific to macrophages, such as the exaggerated proinflammatory responses seen in CGD patients[36], may be a more prominent phenotype in CYBC1-deficient individuals. The complete absence of *gp91$^{phox}$* from macrophages is also in line with some of the infections observed in the p.Tyr2Ter homozygous individuals, like *M.tuberculosis*[37], *Legionella*[38], and *C. albicans*[39], as well as the increased susceptibility of *bc017643*-knockout mice to *S.Typhimurium* and *L.monocytogenes* infections[14]. CYBC1 deficiency thus appears to predispose to a distinct spectrum of microbes that can be considered more specific to macrophages. Supportive of this, a prior study of two distinct *CYBB* mutations causing infections restricted to mycobacteria demonstrated an impaired oxidative burst in macrophages but not in neutrophils[40].

We expect *CYBC1* p.Tyr2Ter to have reached its frequency in Iceland (carried by about 1 in 70) through the founder effect[8]. The founder effect allows such mutations to reach a considerable frequency in the bottlenecked/founder population of Iceland, resulting in homozygotes in high-enough numbers to confer statistical power for discovery of disease associations[8,41]. We have identified eight p.Tyr2Ter homozygous Icelanders in our dataset, comprising 155K chip-genotyped individuals, and would expect 19 homozygotes in the entire Icelandic population (338K individuals). Overall, we observe thirteen Icelanders with a genotype causing signs compatible with CGD, eight homozygous for *CYBC1* p.Tyr2Ter and five for *CYBA* p.Arg90Trp. This amounts to a predicted disease prevalence of 1 in 12,000 in Iceland, much higher than reported for CGD in the literature[2]. In addition to the founder effect in Iceland, this higher overall prevalence of CGD may be explained by the fact that in some instances the causative genotypes lead to a mild phenotype resulting in underdiagnosis of the disease. In Iceland however, we are simply assessing prevalence by identifying pathogenic genotypes.

Our results indicate that CYBC1-deficient individuals have some residual ROS production, similar to what has been reported for CGD patients with *NCF1* biallelic mutations[42]. Residual ROS production has been shown to correlate with a milder clinical presentation, and improved overall survival in CGD depending on early detection and treatment of the disease[42]. Our findings therefore underline the value of using genotypes for the identification of previously undiagnosed cases. Out of the 13 Icelanders with a CGD genotype, only the two probands and two of the *CYBA* p.Arg90Trp homozygotes were, to our knowledge, diagnosed with CGD prior to the discovery of *CYBC1*. Reporting of the genotypes of the remaining nine individuals to their treating physicians is already proving important for their clinical follow-up and treatment.

The set of 155K chip-genotyped Icelanders used for our study consists of individuals healthy enough to have survived up to the time of recruitment, with the vast majority over 18 years old at the time of sample acquisition (mean age at recruitment is 46 years). The set is therefore biased toward adults, and the identification of genotypes from such a set can be expected to reveal greater phenotypic variability than recruitment based on clinical presentation alone, as is illustrated by our study. The mean age at onset of symptoms for the five p.Tyr2Ter homozygous

individuals who had gastrointestinal manifestations is 12.2 years, and deviates somewhat from X-linked recessive CGD patients, the vast majority of whom are diagnosed before 5 years of age[2]. The *CYBC1* p.Tyr2Ter homozygotes also contrast patients with *NCF4* biallelic mutations in terms of loss-of-neutrophil NADPH oxidase activity. The CYBC1-deficient individuals have strongly impaired PMA-induced neutrophil oxidative burst (stimulation indices lower than 3, as in X-linked CGD[43,44], see Fig. 1b and Fig. 2b), whereas NCF4-deficient patients have normal or mildly impaired PMA-induced neutrophil oxidation[5,6]. We recognize that for full understanding of the molecular mechanism of CYBC1 deficiency, a more detailed analysis of NADPH oxidase activity is required, such as assessment of particle-induced ROS production, as well as experiments with other cell types.

Previously, we and others have identified and described a large set of rare complete human knockouts[45–48]. By systematically investigating homozygous loss-of-function carriers, we are beginning to understand how some of these mutations relate to diseases and other traits. Through genotype-based recall of individuals homozygous for *CYBC1* p.Tyr2Ter, we were able to confirm the loss-of-function effect of p.Tyr2Ter by verifying the absence of the CYBC1 protein. Moreover, this allowed us to demonstrate that complete loss of CYBC1 leads to a reduced neutrophil oxidative burst in the absence of pathogenic mutations in known components of the NADPH oxidase complex. Hitherto, pathogenic mutations in five genes encoding subunits of the phagocytic NADPH oxidase complex were known to cause CGD. We have shown that *CYBC1* represents a novel player in the NADPH pathway and the sixth CGD gene.

## Methods

**Study population and generation of genetic dataset**. Our approach to whole-genome sequencing, genotyping, long-range phasing, and imputation has been described in detail in previous publications[8,9]. In brief, 28,075 Icelanders participating in various disease projects at deCODE genetics were whole-genome sequenced using standard TrueSeq methodology (Illumina) to a median depth of 37×. Further, 155,250 Icelanders have been genotyped with Illumina microarrays (chip-genotyped), and genotype probabilites for untyped relatives have been calculated based on Icelandic genealogy. Genotypes of sequence variants identified through sequencing (SNPs and indels) have been imputed into all chip-typed Icelanders and their close relatives (familial imputation).

An extension of this learning set of 28,075 WGS Icelanders are 9185 Icelanders more recently recruited and sequenced, amounting to a set of 37,260 WGS Icelanders used for the current study. These 37,260 Icelanders were all sequenced by the same method as previously described, to a median depth of 38×[8,9].

All participating individuals who donated blood or buccal tissue samples, or their guardians, provided written informed consent. All sample identifiers were encrypted in accordance with the regulations of the Icelandic Data Protection Authority. Personal identities of the participants and biological samples were encrypted by a third-party system approved and monitored by the Icelandic Data Protection Authority. Years of births/deaths in Table 1 of individuals not recruited through clinical sequencing (individuals C–H) are rounded to five years (pedigrees are provided in Supplementary Fig. 6). The study was approved by the Data Protection Authority (ref. 2013030423/ÞS/--, with amendments) and the National Bioethics Committee (ref. VSN 12–121, VSNb2012070013/03.12), that also reviewed and approved the protocol, methodology and all documents presented to the participants. All methods were performed in accordance with the relevant guidelines and regulations.

**Imputation**. Imputation is a method for inferring genotypic status of variants that have not been directly genotyped. Imputation of untyped variants into the mix of typed variants is now routine in human genetics[8]. The extensive genealogical information available for the Icelandic population, along with deCODE's long-range phasing of a large set of genotyped Icelanders[8,49] has increased imputation accuracy and speed by removing uncertainty in phasing. In general, a haplotype can be imputed into an untyped individual if two genotyped relatives share a long haplotype that is identical by descent (IBD), and the genealogy indicates that the path of IBD sharing goes through the individual[49].

The informativeness of genotype imputation is estimated by the ratio of the variance of imputed expected allele counts and the variance of the actual allele counts:

$$\frac{\mathrm{Var}(\mathrm{E}(\theta|\text{chip data}))}{\mathrm{Var}(\theta)},$$

where $\theta$ is the haplotype allele count. Var(E($\theta$ | chip data)) was estimated by the observed variance of the allele imputations and Var($\theta$) was estimated by $p(1\text{-}p)$, where $p$ is the allele frequency. This metric has the property of being 0 when the imputation is completely uninformative (the same value is always imputed) and 1 when the imputation is fully informative. Asymptotically, the expectation of the metric equals the $r^2$ between the imputed alleles and the true genotypes.

**Whole-genome sequencing of the two probands and their family**. Genomic DNA (gDNA) was isolated from blood samples taken pre-HSCT from the two affected brothers and from buccal samples taken in 2016 from their parents and three siblings. A request for whole-genome sequencing (WGS) was sent through an in-house laboratory information management system (LIMS). gDNA was registered and isolated by an in-house core facility (Biological Materials facility). Samples were delivered to the Genome Sequencing Laboratory in a barcoded 96-well tray format and stored at 4 °C until use. Samples were prepared for sequencing using the TruSeq PCR-free library preparation kits from Illumina, and sequenced on Illumina instruments to an average genome-wide coverage of 39×. Details of the sample preparation, paired-end sequencing, read processing and alignment and filtering of resulting BAM files have been described in previous publications[8,9].

**Variant calling, annotation, and filtering**. Variants were called using version 2.3–9 of the Genome Analysis Toolkit (GATK)[50], reads were called with GATK's HaplotypeCaller, version 2014.4-3.3.0-0-ga3711aa, using joint calling. In our analysis of the WGS data of the two probands we focused on SNPs and small indels (shorter than 20 base pairs) at coding and splicing regions in their genomes, as annotated by release 80 of the Variant Effect Predictor[51], using RefSeq gene annotations. We assessed the frequency of observed sequence variants using (1) our set of 37,260 whole-genome sequenced Icelanders[8,9] at a median depth of 38× and gnomAD[32], and (2) 155,250 chip-genotyped Icelanders with 32.5 million imputed variants. We defined rare autosomal recessive genotypes as homozygous or compound heterozygous sequence variants, each with a minor allele frequency lower than 2% in our set and gnomAD[32]. We defined rare autosomal dominant genotypes as variants with a minor allele frequency lower than 0.05% in our set and gnomAD.

**Association testing**. We tested for association between sequence variants and phenotypes under a recessive mode of inheritance, as previously described[52,53]. We used logistic regression to test for association with binary phenotypes, adjusting for sex, age, and county within Iceland. Quantitative phenotypes were tested using a linear mixed model implemented in BOLT-LMM[54]. Height measurements were corrected for year of birth, sex, and age at measurement, using a rank-based inverse normal standardization. A total of 31.6 million variants were tested for association under a multiplicative model. For the recessive analysis, there were 19.2 million variants with homozygotes for the minor allele in the dataset. All of the variants that were tested had imputation information over 0.8. The threshold for genome-wide significance was corrected for multiple testing with a weighted Bonferroni adjustment using as weights the enrichment of variant classes with predicted functional impact among association signals[55]. The significance threshold then becomes $2.5 \times 10^{-7}$ for high-impact variants (including stop-gained, frameshift, splice acceptor or donor), $5.0 \times 10^{-8}$ for moderate-impact variants (including missense, splice-region variants and in-frame indels), $4.5 \times 10^{-9}$ for low-impact variants, $2.3 \times 10^{-9}$ for DNase I hypersensitivity site (DHS) variants and $7.5 \times 10^{-10}$ for remaining variants.

**Phenotypic information**. We performed phenotypic assessment of the eight individuals homozygous for *CYBC1* p.Tyr2Ter by (1) examining a large set of existing clinical, biological, and genealogical information and (2) obtaining novel phenotypic information through an extensive health study of the Icelandic population. The study consists of a 4-h visit with a medical examination, interview and collection of biological samples. For individuals participating in the study we are able to review, through a two-way encryption system, hospital-based information, including hospital discharge diagnoses and laboratory tests. This study allows us to perform deeper phenotyping for individuals targeted by their genotype (genotype-based recall). We have attempted genotype-based recall for a set of ~13,000 loss-of-function homozygous individuals, based on WGS and imputation, and controls[45].

**Sanger sequencing**. The *CYBC1* p.Tyr2Ter mutation was submitted to Sanger sequencing for confirmation of a homozygous genotype in the eight individuals detected by WGS and chip-genotyping. The position of the p.Tyr2Ter mutation was also sequenced by the Sanger method in the five family members of the two probands (Fig. 1a). The *CYBA* p.Arg90Trp mutation was also submitted to Sanger sequencing for all eight *CYBC1* p.Tyr2Ter homozygous individuals. Primers for Sanger sequencing were designed using the Primer 3 software[56]. Following PCR,

cycle sequencing reactions were performed in both directions on MJ Research PTC-225 thermal cyclers, using the BigDye Terminator Cycle Sequencing Kit v3.1 (Life Technologies) and Ampure XP and CleanSeq kits (Agencourt) for cleanup of the PCR products and cycle sequencing reactions. Sequencing products were loaded onto the 3730 XL DNA Analyzer (Applied Biosystems) and analyzed with the Sequencher 5.0 software (GeneCodes Corporation).

**Neutrophil oxidative burst in homozygous individual D**. We performed a neutrophil oxidative burst test for *CYBC1* p.Tyr2Ter homozygous individual D (Table 1). The neutrophil oxidative burst was measured on a fresh whole-blood sample from individual D with a Phagoburst kit (BD biosciences, #341058) used according to the manufacturer's instructions. In short, a venous blood sample was collected in a lithium heparin tube (Vacuette, #455084) and thereafter 100 μL of blood were aliqoted and placed on ice for 10 min. Samples were then stimulated with PMA for exactly 10 min at 37 °C, followed by 10 min incubation with the fluorogenic substrate dihydrorhodamine (DHR) 123 at 37 °C, cell lysis and fixation. Samples were analyzed on a flow cytometer (FACSCalibur, BD and Attune Nxt, Thermo Fisher). Gating strategy is shown in Supplementary Fig. 7.

**Blood cell isolation and macrophage differentiation**. For neutrophil isolation, fresh whole blood was mixed with Lympholyte-poly (Cedarlane #CL5070) at a 1:1 ratio, following the manufacturer's instructions. Thereafter, the sample was centrifuged at 500×g for 35 min and the polymorphonuclear cell layer was extracted by pipetting and resuspended in PBS + 2%FBS. For PBMC isolation, 25 mL of fresh whole blood was added to 15 mL Ficoll-Paque PLUS (GE Healthcare, #17-1440-03) in a SepMate tube (STEMCELL Technologies, #15450). Then, the sample was centrifuged at 1200×g for 10 min. The plasma layer was discarded and the PBMC layer transferred to a fresh tube followed by washes in PBS + 2% FBS.

Untouched monocytes were purified from PBMC preparations using the Monocyte Isolation Kit II (Miltenyi, #130-091-153) following the manufacturer's instructions. Labeled samples were run on MS columns, and both the flow-through (consisting of untouched monocytes), and column bound lymphocytes (mainly B-, T-, and NK-cells) were collected as well for western blot analysis.

Monocyte-derived macrophages were generated by culturing monocytes for 6 days in X-vivo medium (Lonza, #04744Q) supplemented with Glutamax (Thermo Fisher, #35050038), Sodium Pyruvate (Thermo Fisher, #11360039), 50 ng/mL M-CSF (Miltenyi, #130-091-153). Fresh medium was added to cells on day 3 and on day 5 media were aspirated and replaced with the same media with 50 ng/mL IFN-g (Thermo Fisher, #PHC4031). Cells were collected on day 6 for western blot analysis.

**Western blot analysis**. Cell pellets were lysed in 100 μL RIPA buffer (150 mM NaCl, 1% Triton X-100, 0.5% sodiumdeoxycholate, 0.1% SDS, 50 mM Tris, pH 8) with halt protease and phosphatase inhibitor (Thermo Fisher, #78445) and incubated for 10 min on ice. Following brief sonication, samples were spun at 14,000×g for 15 min at 4 °C. The supernatant was transferred to a fresh tube and the protein content of lysates was determined by BCA assay (Thermo Fisher, #23227). Samples were mixed with NuPAGE LDS sample buffer (Thermo Fisher, #B0007) and reducing agent (Thermo Fisher, #B0009), heated for 10 min at 70 °C and loaded on a 4–12% BT bolt plus gel (Thermo Fisher, #NW04120BOX). The gel was equilibrated for 5 min in 20% EtOH and transferred onto a nitrocellulose membrane (Thermo Fisher, # IB23002) with iBlot 2 (Thermo Fisher) running the P0 program. Following the transfer, membranes were re-hydrated in PBS for 5 min and after that blocked in Odyssey blocking buffer (LI-COR, #92740000). Blots were incubated with primary antibodies overnight at 4 °C in Odyssey blocking buffer + 0.05% Tween 20. Antibodies used were, anti-gp91phox at 1:1000 (Santa Cruz, #sc-130543), anti-C17orf62 at 1:1000 (Atlas Antibodies, #HPA045696), anti-beta-actin at 1:2500 (Biotechne, #MAB8929), anti-beta-actin at 1:5000 (Cell Signaling, #4970), anti-mouse IgG (H+L) IRDye 800CW at 1:10,000 (LI-COR, #926–32212), anti-rabbit IgG (H+L) IRDye680RD at 1:10,000 (LI-COR, #926–68073), and anti-rabbit IRDye 800CW at 1:10,000 (LI-COR, #925–32211). Membranes were scanned with Odyssey CLx and subsequent analysis performed in the image studio software (LI-COR). Uncropped blots are provided as Supplementary Fig. 8 and 9.

## Data availability

Our previously described Icelandic population whole-genome sequence data[9] have been deposited at the European Variant Archive under accession PRJEB15197. The authors declare that the data supporting the findings of this study are available within the article, its supplementary information files and upon request. The HGNC accepted our application for the gene symbol *CYBC1* instead of the placeholder *C17orf62* on February 26th 2018, listed under accession 28672.

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

## Acknowledgements

We wish to thank the family of the two probands, as well as all the other individuals who participated in the study and whose contribution made this work possible.

## Author contributions

G.A.A., G.L.N., S.S., U.T., I.J., P.S., and K.S. designed the study and interpreted the results. G.L.N., S.G., A.B.A., K.B., F.T., and P.M. performed laboratory experiments and interpreted the results. S.A.G., G.M., G.A.T., and D.F.G. developed the methods for data analysis. J.S., K.B.O., H.H., Si.B., R.A., and A.H. carried out or supervised phenotyping. O.Th.M., Ad.J., As.J., and A.S. performed the sequencing. G.A.A., B.O.J., S.B., E.V.I., A. O., R.P.K., G.S., K.F.A., T.J., K.R.G., D.F.G., and P.S. performed the statistical and bioinformatics analyses. G.A.A., G.L.N., D.F.G., U.T., I.J., A.H., P.S., and K.S. drafted the manuscript. All authors contributed to the final version of the paper.

## Additional information

**Competing interests:** The authors affiliated with deCODE genetics/Amgen declare a conflict of interest as employees. The remaining authors declare no competing interests.

