## [Peer Review File · Nature Communications]

Reviewer #1 (Remarks to the Author):

This interesting study describes the first human patients who have a homozygous loss of function mutation in the gene encoding EROS. EROS is a chaperone important for ER assembly of the flavocytochrome heterodimer, the electron transferase in phagocyte NADPH oxidase. This enzyme generates superoxide, the precursor to potent oxidants required for killing of certain bacteria and fungi. The function of EROS was recently discovered and characterized in mice following an unbiased assessment of various KO mice for susceptibility to infections (as noted in the cited reference, Thomas DC et al JEM 2017).

Inactivating recessive mutations in any one of 4 enzyme subunits of the NADPH oxidase result in loss of enzyme activity and the primary immunodeficiency disorder chronic granulomatous disease (CGD). These include the 2 subunits of flavocytochrome b and two regulatory proteins. CGD is characterized by recurrent severe bacterial and fungal infections involving skin/soft tissue and their draining lymph nodes, lung, and abscesses in the liver, bone, or brain. A distinctive spectrum of microbes cause infections in CGD. In addition, patients have inflammatory manifestations including granulomatous inflammation in the GI tract that resembles Crohn's disease. This is believed to reflect the influence of NADPH oxidase ROS on regulating inflammatory signaling and other processes not directly related to their microbicidal effect. Recessive loss of function mutations in a fifth regulatory subunit, leading to a partial loss of oxidase activity, is thus far described only in one patient who had GI disease, but no deep-tissue or lung infections.

Defects in EROS were identified by WGS in two Icelandic brothers who had granulomatous GI disease and bacterial infections including mouth ulcers that cultured positive for *B. cepacia*. As *B. Cepacia* is highly suggestive of underlying CGD, neutrophil NADPH oxidase assays (DHR flow cytometry) were done and revealed substantially reduced oxidase activity. The subsequently identified mutation in EROS predicts a premature stop codon near the N-terminus of the protein that results in loss of EROS expression. The prior study on EROS KO mice showed a marked decrease in flavocytochrome b in PMNs and in macrophages w / substantially reduced oxidase activity, which was relatively preserved, however, in response to the stimulus SOZ (serum opsonized zymosan).

The identified EROS allele appears to be a very rare variant distributed in Iceland due to a founder effect. Subsequent studies in the Icelandic population identified six additional homozygotes, 3 of who had inflammatory bowel disease. Some patients also had infections described although not all are characteristic of CGD (e.g. invasive pneumococcal disease, viral infections).

The study presents novel findings that are of interest, and expand the scope of defects causing human innate immune defects. In particular, this study for the first time reports the impact of a gene

defect in a protein that important for the biosynthesis of a crucial NADPH oxidase subunit. The genetic analysis appears solid, but more work is needed on characterizing the impact of the loss of EROS on phagocyte function and the clinical phenotype, as well as a more nuanced discussion. From my perspective, it appears that loss of EROS results in a partial NADPH oxidase defect with a variable phenotype that had severe manifestations in one kindred (two brothers who had both recurrent bacterial infections as well as granulomatous GI disease) but was often milder or absent in others, with GI symptoms being the most consistent feature.

Specific comments

1. Characterization of the clinical phenotype:

It should be noted that most patients are atypical of CGD in that they had a limited number of the bacterial infections characteristic of CGD, and also did not have *Aspergillus*, which is far more common than *Candida* in CGD - 50% of patients have at least one *Aspergillus* infection, typically lung, and this is the major cause of morbidity and mortality in CGD (B Marciano et al 2015). The *Burkholderia cepacia* was identified only in a mouth ulcer, which is also atypical; it is described as a cause of pneumonia, lymphadenitis, and (often fatal) bacteremia in CGD. As already mentioned, invasive pneumococcal disease and viral infections are not part of the CGD phenotype.

More information is needed on the GI disease in patient E and F. Are there biopsy samples showing granulomatous inflammation? This is important. "Ulcerative colitis" is not a feature of CGD. It is also helpful to have specific information on patient C.

For patient G, interstitial pulmonary disease with fibrosis in the absence of prior pneumonia is not typical of CGD, in my opinion.

More clinical information on the index patients – it is stated that their "condition deteriorated", thus leading to hematopoietic stem cell transplantation. Please explain, was this clinical deterioration due to difficult to manage GI disease?

3. The discussion on the mouse cellular phenotype is very brief and needs to be expanded just a bit. For example, how reduced was the enzyme activity and protein expression and did loss of EROS in the mouse affect both neutrophils and monocyte-macrophages similarly?

4. The oxidase activity is limited only to a DHR assay in three patients (2 brothers from the original kindred and 1 other patient (D)). This assay evaluates intracellular ROS in response to an artificial stimulus, phorbol myristate acetate. I recognize that it may be difficult to perform more detailed

assessment on these patients, eg. a comparison of intracellular and extracellular ROS as elicited by other stimuli. However, the limitations of the DHR assay should be noted. It would be beneficial to study additional patients, as well. The DHR assay can be performed using whole blood shipped to a reference laboratory.

The current depiction of the data shows that the patient PMN cells have some residual NADPH oxidase activity. The presentation and analysis should be modified.

Overlays of the patient cells with their corresponding unstimulated controls and, separately, the paired analysis of the unstimulated and stimulated normal donor controls should be shown rather than as currently depicted. A more quantitative analysis of how reduced the oxidase activity is should be given, for example, as a percent activity of patient relative to the control, based on the MFI of the response, not “% positive” (see ,e.g Vowells et al, <https://www.ncbi.nlm.nih.gov/pubmed/85513>, and <http://www.cytometry.org/newsletters/eICCS-2-1/article6.php>

5. The western blots for the neutrophils and monocyte-derived macrophages should be shown in the body of the paper. These are the relevant cells, not lymphocytes.

The neutrophils from the single patient (D) where western blots were performed show a substantial amount of gp91. It is puzzling that the DHR assay -based oxidase activity was more reduced in this patient , but this may reflect the limitation of the DHR assay. Interestingly, gp91 expression in monocyte derived macrophages from this same patient are much more reduced, almost absent. This suggests that the role of EROS may be more important in this lineage in humans, rather than neutrophils.

6. Residual expression and oxidase activity in PMN may in part explain the limited infections in most of the EROS-deficient patients. Those that had many more infections, such as the two brothers, could reflect the effects of variants in other genes important for innate immune responses or environmental influences.

It would be very helpful to analyze gp91 expression in neutrophils and monocyte derived macrophages on other patients. This could be done by western blot. There is also an antibody for recognition of human CYBB, 7D5, that can be used in flow cytometry. As already mentioned, oxidase assays on additional patients would also be helpful.

7. It is stated in the text that gp91 levels in PMN from patient D are strongly reduced, but gp91 protein is easily seen in the blot in the supplemental figure, where the legend states that gp91 is only 50% reduced. This is a disconnect.

8. HLH is associated with CGD, not necessarily “preceding it”, which is how it is stated on line 186.

Reviewer #2 (Remarks to the Author):

In this manuscript the authors describe patients with homozygous mutations in C17orf62 and a clinical and/or biochemical phenotype consistent with Chronic Granulomatous Disease (CGD). The index cases were two brothers who were clinically suspected of CGD and confirmed by oxidative burst testing. Both were demonstrated to have homozygous mutations in C17orf62 by whole genome sequencing (WGS). Through subsequent analysis of WGS and gene chip data from an Icelandic population, a further six individuals with the same homozygous mutation were identified. Of the 8 patients identified, 7 had clinical features suggestive of CGD and the remaining patient had reduced oxidative burst. Based on their findings, the authors conclude that loss of function mutations in C17orf62 represent a new cause of CGD.

Overall, the findings presented here are important and novel with application for clinical and scientific communities but, in my opinion, require a number of clarifications and further evidence to strengthen the conclusions.

Specific comments:

Further details of genetic analysis should be provided and in particular what the level of coverage was for other CGD genes in the WGS and gene chip cohorts. This is especially important as NCF1 is notoriously poorly covered: a median coverage of 38x of WGS does not ensure that NCF1 is covered so coverage metrics need to be stated explicitly (for other CGD genes as well). All variants in these genes should be presented, along with relaxation of MAF filters in these genes given the fact that the population has a founder effect. How common are EROS homozygous variants in EXAC within and outside Iceland?

There is insufficient clinical and biochemical data provided here for individual patients given that this is intended to describe a new form of CGD. The authors state that they have the ability to recall patients as part of the gene screening study. In my opinion, all patients should have biochemical confirmation of reduced ROS, and ideally reduced gp91Phox. In addition to consolidating the data shown, biochemical confirmation would be a routine part of making a CGD diagnosis.

It would also be useful to know whether there is preservation of a degree of oxidative burst to explain the somewhat milder infection phenotypes described compared with other genetic causes

CGD. For example in Figure 1 there is some evidence of ROS production by the patients- how does this compare with CYBB patients? Is the oxidative burst similarly impaired for all EROS patients or is there variation? How many patients were on immunosuppressive agents at the time that their oxidative burst was done?

Figure 3: Why are the homozygous patients not represented on the quantitative graph should in part a)? The authors should explain how there is absent EROS protein in one patient and 50% EROS expression in heterozygous carriers if NMD is not associated with this mutation.

In supplementary Figure 5 how do the authors explain absence of gp91Phox in Non-carrier D and the discrepancy between neutrophil and macrophage levels of gp91Phox for Homozygote D (Supplemental figure 3)?

More clinical information should be provided especially for types of infection and severity of bowel disease. The infection history shown for the cohort in Table 1 details some organisms and infections that are commonly associated with CGD (eg Burkholderia and disseminated TB) but others that are not, such as viral infections and invasive pneumococcus. There are notable absences for Aspergillus and Staph Aureus infections that are particularly common in other forms of CGD: this should be discussed. Additionally some feature are included in the infection column that would be better included under IBD phenotype (eg anal fistula) and in the immunological/inflammatory column that would be better under infection (eg otitis media). The authors should provide a case history and pedigree for each individual in supplementary data to give a better idea of the clinical spectrum associated with the genotype described. Were there other family members sequenced for pedigrees other than the index pedigree?

In general, the text is poorly focused and should be shortened with concentration on flow of the information presented. There is repetition and a relatively small amount of space is taken describing the main findings about EROS defects. For example, much of the text in lines 187 onwards is repetition. In addition the insertions of text about all forms of CGD in Iceland does not particularly add to the manuscript/ data presented here. For example, lines 197-200. It is not clear here whether the 4 previously diagnosed were all CYBB mutations or whether the index cases here are included. It is also surprising that patients with CYBB mutations were not diagnosed with CGD prior to WGS or chip sequencing as the diagnosis is usually picked up by clinical presentation. I am not sure that this manuscript is the correct place to combine the discussion of CGD in Iceland caused by other known genetic causes as it distracts from the main message.

Information about the Saudi-Arabian patient is interesting but HLH is usually the consequence of infection in CGD. Can the authors be contacted to clarify the clinical phenotype of this patient- for example did the patient have colitis?

Minor comments:

The word 'imputation' is unclear in meaning for a broad audience and should be re-phrased/explained. Similarly, the phrase '(imputation information =112 0.99)' should be explained for a non-genetic audience- or removed if not relevant.

In the abstract, it is not clear whether the following sentence refers to analysis within the EROS mutation cohort or across the whole genome/gene chip data sets: 'Homozygosity for 39 p.Tyr2Ter associates with inflammatory bowel disease (IBD) ($P = 8.3 \times 10^{-8}$; OR (95% CI) = 40 67.6 (14.5, 315.5)), and growth retardation ($P = 3.3 \times 10^{-4}$; Effect (95% CI) = -1.24 SD (-41 1.92, -0.56) / -8.5 cm), consistent with mutations in other known CGD genes'

Lines 71-76 would be better combined with the text prior to description of the index cases (lines 49-59)

Line 93: GCGR should be explained

Line 103/4: the final sentence here is superfluous.

Lines 128/129: 'In patients with CGD, 129 microbial catalase is traditionally recognized as the most important virulence factor¹⁴' should be clarified 'for infection'.

Lines 133-137: This sentence could be removed.

Line 138: How confident are the authors of the clinical phenotype provided for patient G: pulmonary fibrosis is seen in CGD, often as a result of previous infections- did this patient have a history of lung infections? Also pneumococcal disease is not one of the 'common/typical' CGD infections.

Lines 191-193: is this a fair way to estimate prevalence as presumably CGD patients diagnosed by candidate gene sequencing following recognition of the classical clinical phenotype would not then be submitted for WGS/gene chip: could the selection of patients for genetic studies specifically exclude those with known diagnoses?

Line 197: sentence starting 'In particular..': I don't understand this as there are already 5 CYBB patients alluded to in the preceding text.

Figure 3: it would be more intuitive to show homozygote results before heterozygote results.

Supplementary Table 1: I do not think this adds to this paper and I would remove it along with discussion of CYBB patients in the text.

Reviewer #3 (Remarks to the Author):

This manuscript sheds further light into the pathogenesis of CGD, expanding the list of genes implicated in the disease and providing clinical data to support findings in mouse models that have uncovered a new mechanism for CGD.

1) Overall, the impact of the findings could be strengthened with additional functional data on the neutrophils of these homozygous donors, by evaluating microbial killing, NET formation and hyperinflammation (ROS suppresses cytokine production in response to small microbes). These data would help solidify the functional association between the defects in ROS burst and the clinical pathology (recurrent infections, hyperinflammation) reported for these individuals.

2) Pg. 3 In 70. It would be useful to know how long the BM transplant recipient has been symptom free.

Reviewer #4 (Remarks to the Author):

The manuscript by Arnadotti et al. reports a new disease-causing gene for CGD, EROS, which was identified in 8 patients from Iceland diagnosed with CGD. These patients were homozygous for a loss-of-function mutation, p.Tyr2Ter, which produces truncated protein. This variant is relatively common (MAF= 0.76%) in the Iceland population and absent in the gnomAD database. Heterozygosity for this variant does not cause a decrease in the EROS mRNA expression, although the protein expression appears to be reduced by 50%. Individuals homozygous for this variant have null protein expression. Together, these data provide solid evidence that this variant is pathogenic.

In addition to this variant being linked to CGD, it is strongly associated with IBD under the recessive model of inheritance with the impressive odds ratio OR=67.6. Homozygosity for this mutation is also associated with reduced height in the same population, which could be reflection of the inflammatory phenotype in these individuals.

The authors showed absence of mutant EROS protein in lymphocytes of two patients, decreased oxidative burst in neutrophils of 3 patients including the asymptomatic individual D. Protein expression of the known CGD gene that is regulated by EROS, gp91 phox, was reduced in primary cells of homozygous individuals (one affected and one unaffected). These functional experiments further support the genetic evidence that LOF mutations in EROS are linked to CGD and/or IBD. Mice deficient for Eros are susceptible to inflammation which is consistent with the infectious phenotype in patients with CGD.

In summary, this is a fairly straight forward genetic study that has benefited from the exquisite data collection (genetic and medical records) in the Iceland population.

Comments

Genetics:

1. Are there other genes in the index family that passed the filtering criteria beside the two reported genes, GCGR and EROS? If so, please show them in the supplementary table. As the filtering at allele freq. <2% is not very stringent, although it is acceptable for studies in founder populations, it is possible that these two patients may have pathogenic mutations in other genes that are linked to immunodeficiency (typically recessively inherited disorders and more common in founder populations).
2. Please discuss why the individual D have no disease features despite being homozygous for the variant and having abnormal neutrophil oxidative burst test. Are there for example variants in her exome that may function as modifiers or protective variants for IBD? Are there protective variants for IBD from GWAS studies?
3. The allele frequency for the Tyr2Ter mutation is almost 1% in this population (q2 is 1:10,000) and 8 homozygous patients with likely CGD phenotype were identified in a cohort of about 180,000 people. However, the total number of homozygous individuals in the entire data set of ~340,000 people is unclear. Homozygosity for this variant is strongly associated with IBD and reduced height, which suggests that there could be more homozygous individuals in the studied population. The association data for IBD and reduced height should be shown in table format i.e. allele frequencies in patient and control cohorts. Please clarify whether the 8 patients with CGD are also included in these association studies. Most GWAS-identified variants for IBD have OR in the range of 1-3, which raises the question whether the EROS gene is actually a new IBD recessively inherited gene. In support of this hypothesis is the fact that 5/7 symptomatic patients diagnosed with CGD have IBD and that infectious history is not very strong in all 8 patients (patients C, E, F have none or viral infections) (table 1). Patients A and B have strong clinical history and functional evidence (burst test) for CGD while other patients are less convincing. This is understandable as their clinical data was not complete and is based on self-reported medical records. Please comment.

Functional studies:

Although several experiments showed the effect of p.Tyr2Ter on the protein expression and function, this manuscript is not very strong with functional data.

The oxidative burst experiments are fairly convincing. There is some ambiguity regarding the mRNA expression and protein expression of mutant EROS. NMD bypass of protein truncating mutations that are close to the start codon may explain normal mRNA expression in carriers for p.Tyr2Ter (Lindeboom et al, 2017). Despite no difference in the mRNA or allelic expression of mutant EROS in the heterozygous carriers vs non-carriers, the protein expression is decreased in mutation carriers based on the quantification of western blots, which are otherwise not convincing (supplementary figure 4). Could there be an alternatively translated EROS protein as the result of downstream re-initiation in translation? Does anti-C17orf62Ab detect the N-terminal or the C-terminal end of the protein?

What is the mRNA expression in homozygous individual F or D? (their primary cells were available for other experiments) Does mRNA expression in homozygous individuals correlate with the absent mutant protein?

What is the explanation for inconsistency in Gp91phox protein expression in the primary cells of individual D as shown in sup. fig 3. and 5 (50% reduction vs. null expression). Both experiments were done in myeloid cells that highly express EROS and Gp91phox, thus this discrepancy is unlikely due to cell-type specific expression. It also appears that the same Gp91phox antibody was used for both western blots.

This study is a bit short of details on the function of human EROS protein, which is still largely unknown. The authors discussed and speculated its function based on the murine model of Eros deficiency. Primary patients cells are available to replicate some experiments from the murine model study. What is the mechanism by which EROS regulates the expression of gp91phox and gp22phox? Studies in KO mice suggested that Eros might regulate the proteosomal degradation of gp91 phox and gp22phox. Is that the case in human cells? Readers of this journal would appreciate more discussion and data on the EROS function in humans.

Other suggestions:

Figure 1: please add the panel showing EROS protein structure with domains and the location of the causal variant.

Figure 2: not informative and it should be moved to supplementary. It could be replaced with the table showing data from association studies.

Author's response to reviewers' comments

Gene nomenclature

In our initial submission, we had followed the nomenclature from the recent publication of the murine ortholog of *C17orf62*, i.e. *Eros* (Thomas et al., 2017). Our application to the HUGO gene nomenclature committee (HGNC) for renaming of *C17orf62* as *EROS* was declined. After discussion with the HGNC the genename *CYBC1* has been chosen and formally accepted based on the following rationale:

1. *CYBC1*; Cytochrome b-245 Chaperone protein 1. Since the animal and human experiments point to the protein encoded by *C17orf62* being a chaperone protein for the dimerization of cytochrome b-245 (gp91phox and p22phox).
2. The *CYBC1* name follows convention within the previously defined group of genes mutated in chronic granulomatous disease patients (*CYBA* and *CYBB*).

URL: https://www.genenames.org/cgi-bin/gene_symbol_report?hgnc_id=HGNC:28672

From now on, in our revised manuscript and in the future, we will adhere to the name *CYBC1* for *C17orf62*.

Reviewers' comments:

Reviewer #1 (Remarks to the Author):

This interesting study describes the first human patients who have a homozygous loss of function mutation in the gene encoding EROS. EROS is a chaperone important for ER assembly of the flavocytochrome heterodimer, the electron transferase in phagocyte NADPH oxidase. This enzyme generates superoxide, the precursor to potent oxidants required for killing of certain bacteria and fungi. The function of EROS was recently discovered and characterized in mice following an unbiased assessment of various KO mice for susceptibility to infections (as noted in the cited reference, Thomas DC et al JEM 2017).

Inactivating recessive mutations in any one of 4 enzyme subunits of the NADPH oxidase result in loss of enzyme activity and the primary immunodeficiency disorder chronic granulomatous disease (CGD). These include the 2 subunits of flavocytochrome b and two regulatory proteins. CGD is characterized by recurrent severe bacterial and fungal infections involving skin/soft tissue and their draining lymph nodes, lung, and abscesses in the liver, bone, or brain. A distinctive spectrum of microbes cause infections in CGD. In addition, patients have inflammatory manifestations including granulomatous inflammation in the GI tract that resembles Crohn's disease. This is believed to reflect the influence of NADPH oxidase ROS on regulating inflammatory signaling and other processes not directly related to their microbicidal effect. Recessive loss of function mutations in a fifth regulatory subunit, leading to a partial loss of oxidase activity, is thus far described only in one patient who had GI disease, but no deep-tissue or lung infections.

Defects in EROS were identified by WGS in two Icelandic brothers who had granulomatous GI disease and bacterial infections including mouth ulcers that cultured positive for *B. cepacia*. As *B. Cepacia* is highly suggestive of underlying CGD, neutrophil NADPH oxidase assays (DHR flow cytometry) were done and revealed substantially reduced oxidase activity. The subsequently identified mutation in EROS predicts a premature stop codon near the N-terminus of the protein that results in loss of EROS expression. The prior study on EROS KO mice showed a marked decrease in flavocytochrome b in PMNs and in macrophages w / substantially reduced oxidase activity, which was relatively preserved, however, in response to the stimulus SOZ (serum opsonized zymosan).

The identified EROS allele appears to be a very rare variant distributed in Iceland due to a founder effect. Subsequent studies in the Icelandic population identified six additional homozygotes, 3 of who had inflammatory bowel disease. Some patients also had infections described although not all are characteristic of CGD (e.g. invasive pneumococcal disease, viral infections).

The study presents novel findings that are of interest, and expand the scope of defects causing human innate immune defects. In particular, this study for the first time reports the impact of a gene defect in a protein that important for the biosynthesis of a crucial NADPH oxidase subunit. The genetic analysis appears solid, but more work is needed on

characterizing the impact of the loss of EROS on phagocyte function and the clinical phenotype, as well as a more nuanced discussion. From my perspective, it appears that loss of EROS results in a partial NADPH oxidase defect with a variable phenotype that had severe manifestations in one kindred (two brothers who had both recurrent bacterial infections as well as granulomatous GI disease) but was often milder or absent in others, with GI symptoms being the most consistent feature.

Specific comments

I. Characterization of the clinical phenotype:

I.a. It should be noted that most patients are atypical of CGD in that they had a limited number of the bacterial infections characteristic of CGD, and also did not have *Aspergillus*, which is far more common than *Candida* in CGD - 50% of patients have at least one *Aspergillus* infection, typically lung, and this is the major cause of morbidity and mortality in CGD (B Marciano et al 2015). The *Burkholderia cepacia* was identified only in a mouth ulcer, which is also atypical; it is described as a cause of pneumonia, lymphadenitis, and (often fatal) bacteremia in CGD. As already mentioned, invasive pneumococcal disease and viral infections are not part of the CGD phenotype.

Answer: Firstly, we would like to note that both phenotypic variability and variable expressivity are accepted for the other known CGD genes.

Secondly, we have observed and agree with the reviewer that not all eight patients we describe have the same presentation of the disease, in terms of the number and severity of infections. However, we believe this can be partly explained by the unique design of our study. The two most severely affected individuals were identified from their clinical presentation, whereas the other six were identified on the basis of their genotype (since we have genotypic information for 155K Icelanders). The fact that our patient group has less severe/typical CGD phenotypes is therefore not surprising. Rather, it could be argued that earlier studies of the genetics of CGD had an ascertainment bias, since they only included individuals with an extreme phenotypic presentation.

We consider it likely that individuals exist who have not been diagnosed with CGD, possibly due to mild affection status, despite carrying a known pathogenic CGD genotype (mutations in *CYBA*, *CYBB*, *NCF1*, *NCF2*, or *NCF4*). These individuals would show an abnormal oxidative burst if tested.

We have added a sentence to the Discussion section addressing this (page 12, line 256):

„Notably, recruitment based on genotypes can be expected to reveal greater phenotypic variability than recruitment based on clinical presentation alone, as is illustrated by our study.“

I.b. More information is needed on the GI disease in patient E and F. Are there biopsy samples showing granulomatous inflammation? This is important. “Ulcerative colitis” is not a feature of CGD. It is also helpful to have specific information on patient C.

Answer: In response to the reviewer's request we have gathered more detailed clinical information on the gastrointestinal symptoms of patients E, F, and C, from their gastroenterologists. We have now extended the discussion on the GI disease in these patients in the main text as follows (page 6, line 118):

„Of the three additional homozygotes suffering from colitis, two had a confirmed granulomatous colitis (individuals C and F). Individual C was initially diagnosed with ulcerative colitis (UC), receiving a CD diagnosis only after undergoing total colectomy that revealed extensive granulomatous inflammation of the colon. The third individual, individual E, was diagnosed at 14 years of age with total colitis. Repeated biopsies from colon did not yield granulomatous changes. Interestingly, even though granulomas were not observed in individual E, his clinical course could be considered to better fit CD than UC. Notably, granulomatous changes are unconfirmed in the majority of CGD patients with colitis¹⁶. Furthermore, diffuse colitis involving all regions of the colon, as seen for individual E, could be considered characteristic of the gastrointestinal involvement in CGD¹⁶. “

I.c. For patient G, interstitial pulmonary disease with fibrosis in the absence of prior pneumonia is not typical of CGD, in my opinion.

Answer: Individual G likely suffered from pneumonia, she was noted to have invasive pneumococcal disease at 30 years of age caused by *Streptococcus pneumoniae* (we have now added infectious pathogens to Table 1). The *S.pneumoniae* positive culture was taken from blood, but we do not have cultures from sputum and therefore cannot verify whether or not the infectious episode reached the lungs. Our infectious registry dates back to 1985 (at the time of the invasive pneumococcal episode in individual G) but the hospital discharge diagnoses do not.

In response to the reviewer's comment we have modified the sentence discussing interstitial pulmonary disease in patient G. We previously stated that interstitial pulmonary fibrosis is a known manifestation of CGD, but now refer to it as a noted complication due to the exaggerated inflammatory response in CGD (as discussed in Mahdavian et al., 2013). The sentence now reads (page 7, line 147):

“Individual G had invasive pneumococcal disease at 30 years of age caused by *Streptococcus pneumoniae*⁷, and later developed interstitial pulmonary fibrosis, a noted complication of the exaggerated inflammatory response in CGD. “

I.d. More clinical information on the index patients – it is stated that their “condition deteriorated“, thus leading to hematopoietic stem cell transplantation. Please explain, was this clinical deterioration due to difficult to manage GI disease?

Answer: According to the index patients' (brothers') treating pediatrician they had gone through repeated periods of steroid treatment directed at their severe gastrointestinal symptoms, with only limited clinical benefit. This, along with the fact that the older brother developed an enlarged liver and spleen, led to the decision of the haematopoietic stem cell transplantation. We have now changed the clinical description (in the Supplementary Material) regarding their deterioration as follows:

„In the following months, it became increasingly difficult to treat the brothers for gastrointestinal symptoms. A treatment of repeated periods with steroids showed only limited and temporary improvement. The condition of older brother, who also developed an enlarged liver and spleen, was particularly severe. Due to the progression of their disease, it was decided that the brothers would undergo haematopoietic stem cell transplantation (HSCT) in Newcastle, both from unrelated donors.“

2.-3. The discussion on the mouse cellular phenotype is very brief and needs to be expanded just a bit. For example, how reduced was the enzyme activity and protein expression and did loss of EROS in the mouse affect both neutrophils and monocyte-macrophages similarly?

Answer: We have now expanded our discussion on bc01743(Eros)-knockout mice in the main text. We have modified the section when first introducing the C17orf62 genotype, to include a discussion on the impairment of the oxidative burst in both neutrophils and macrophages from the knockout mice (page 5, line 97):

„C17orf62 shares 89% amino acid sequence identity with the murine bc017643 which was recently shown to be essential for ROS production², consistent with the reduced ROS production described for CGD patients. Both neutrophils and BM-derived macrophages from bc017643-knockout mice showed a highly impaired oxidative burst in response to a range of stimuli, including PMA. Furthermore, the ROS deficit observed in macrophages was even more pronounced than in neutrophils, resembling those of gp91^{phox}-deficient mice. Accordingly, bc017643-knockout mice (-/-) show a phenotype similar to mice deficient in individual subunits of the NADPH oxidase, including high susceptibility to infections².“

We have further extended the discussion on bc01743-knockout mice to include the expression changes of all protein subunits of NADPH in mouse neutrophils and macrophages (page 9, line 193):

„Neutrophils and bone marrow (BM)-derived macrophages from bc017643-knockout mice had markedly reduced expression of both gp91^{phox} and p22^{phox}, despite expressing normal mRNA levels for these proteins, most likely due to the role of bc017643 as a chaperone in the dimerization of cytochrome b-245. The cytosolic components of NADPH oxidase (p47^{phox}, p67^{phox}, and p40^{phox}) were not differentially expressed in bc017643-deficient mice².“

4.a. The oxidase activity is limited only to a DHR assay in three patients (2 brothers from the original kindred and 1 other patient (D)). This assay evaluates intracellular ROS in response to an artificial stimulus, phorbol myristate acetate. I recognize that it may be difficult to perform more detailed assessment on these patients, eg. a comparison of intracellular and extracellular ROS as elicited by other stimuli. However, the limitations of the DHR assay should be noted. It would be beneficial to study additional patients, as well. The DHR assay can be performed using whole blood shipped to a reference laboratory.

Answer: We agree with the reviewer that we would ideally have preferred to study all eight p.Tyr2Ter homozygous individuals. However, out of the eight, two are deceased, and one had

bone-marrow transplantation. Of the remaining five, three were contacted and were unable to come for further analysis at the phenotyping center, two came (of whom only one was able to come a second time for acquisition of a blood sample for fresh neutrophils).

We have now added a sentence stating the limitation of the DHR assay to the Methods section:

“We note that the neutrophil oxidative burst test performed was limited in that it only evaluated intracellular ROS, and only a single, artificial stimulus (PMA) was used.”

4.b. The current depiction of the data shows that the patient PMN cells have some residual NADPH oxidase activity. The presentation and analysis should be modified.

Overlays of the patient cells with their corresponding unstimulated controls and, separately, the paired analysis of the unstimulated and stimulated normal donor controls should be shown rather than as currently depicted. A more quantitative analysis of how reduced the oxidase activity is should be given, for example, as a percent activity of patient relative to the control, based on the MFI of the response, not “% positive” (see ,e.g Vowells et al, <https://www.ncbi.nlm.nih.gov/pubmed/85513>, and <http://www.cytometry.org/newsletters/eICCS-2-1/article6.php>

Answer: We have now modified the burst test figures as suggested by the reviewer (see Figure 1 and Figure 3). In addition, we have now calculated stimulation indices for the three tested homozygotes. As detailed in Vowells et al., the stimulation index (SI) is calculated by dividing the mean fluorescence of DHR in PMA stimulated cells by unstimulated cells. We have added the stimulation indices to the legends of Figures 1 and 2 (showing the oxidative burst test results).

5.a. The western blots for the neutrophils and monocyte-derived macrophages should be shown in the body of the paper. These are the relevant cells, not lymphocytes.

Answer: Following the reviewer’s suggestion we have now moved the western blot for CYBC1 and gp91^{phox} in neutrophils from individual D to the body of the paper (previously Supplementary Figure 3, now part of Figure 2), as well as showing the quantification of the western blot results for gp91^{phox} in monocyte-derived macrophages from individuals D and F (Supplementary Figure 5).

5.b. The neutrophils from the single patient (D) where western blots were performed show a substantial amount of gp91. It is puzzling that the DHR assay -based oxidase activity was more reduced in this patient , but this may reflect the limitation of the DHR assay. Interestingly, gp91 expression in monocyte derived macrophages from this same patient are much more reduced, almost absent. This suggests that the role of EROS may be more important in this lineage in humans, rather than neutrophils.

Answer: Apart from the fact that outcome of the DHR test can be expected to vary somewhat between experiments, our quantification of the neutrophil oxidative burst shows that the the

oxidase activity is not more reduced in individual D ($SI_D=2.35$) than in individuals A and B ($SI_A=1.34$ and $SI_B=2.50$, respectively).

In response to the reviewer's suggestion of a more important role of CYBC1 (EROS) in macrophages than in neutrophils we have now added the following text to the Discussion section (page 11, line 225):

„Although no conclusion can be drawn from only one individual, the difference in expression of gp91^{phox} between monocyte-derived macrophages (absent) and neutrophils (50% reduced) from homozygous individual D is noteworthy, and suggests the presence of more than one chaperone for cytochrome b-245 in the neutrophil lineage.”

6. Residual expression and oxidase activity in PMN may in part explain the limited infections in most of the EROS-deficient patients. Those that had many more infections, such as the two brothers, could reflect the effects of variants in other genes important for innate immune responses or environmental influences.

It would be very helpful to analyze gp91 expression in neutrophils and monocyte derived macrophages on other patients. This could be done by western blot. There is also an antibody for recognition of human CYBB, 7D5, that can be used in flow cytometry. As already mentioned, oxidase assays on additional patients would also be helpful.

Answer: Again, as detailed in our response to comment 4.a. from the reviewer, additional patients (p.Tyr2Ter homozygotes) were unfortunately not available to us for this type of analysis.

7. It is stated in the text that gp91 levels in PMN from patient D are strongly reduced, but gp91 protein is easily seen in the blot in the supplemental figure, where the legend states that gp91 is only 50% reduced. This is a disconnect.

Answer: We have now changed „strongly reduced“ to „an approximately 50% reduction“.

8. HLH is associated with CGD, not necessarily “preceding it”, which is how it is stated on line 186.

Answer: We have now modified the sentence so that it contains more detailed information about the phenotypes reported for the patient from the Saudi-Arabian publication, in response to comment 12 from Reviewer #2 (page 10, line 213):

„Interestingly, the individual homozygous for that mutation had symptoms compatible with CGD, including lymphadenopathy, hepatosplenomegaly, recurrent fever, and growth retardation³⁰“

Reviewer #2 (Remarks to the Author):

In this manuscript the authors describe patients with homozygous mutations in C17orf62 and a clinical and/or biochemical phenotype consistent with Chronic Granulomatous Disease (CGD). The index cases were two brothers who were clinically suspected of CGD and confirmed by oxidative burst testing. Both were demonstrated to have homozygous mutations in C17orf62 by whole genome sequencing (WGS). Through subsequent analysis of WGS and gene chip data from an Icelandic population, a further six individuals with the same homozygous mutation were identified. Of the 8 patients identified, 7 had clinical features suggestive of CGD and the remaining patient had reduced oxidative burst. Based on their findings, the authors conclude that loss of function mutations in C17orf62 represent a new cause of CGD.

Overall, the findings presented here are important and novel with application for clinical and scientific communities but, in my opinion, require a number of clarifications and further evidence to strengthen the conclusions.

Specific comments:

I.a. Further details of genetic analysis should be provided and in particular what the level of coverage was for other CGD genes in the WGS and gene chip cohorts. This is especially important as NCF1 is notoriously poorly covered: a median coverage of 38x of WGS does not ensure that NCF1 is covered so coverage metrics need to be stated explicitly (for other CGD genes as well).

Answer: We thank the reviewer for a well raised point and have now added a coverage table (Supplementary table 2) for the six genes (CYBB, CYBA, NCF1, NCF2, NCF4 and C17orf62) that includes the individual coverage for all eight p.Tyr2Ter homozygous individuals on WGS (all of whom have now been whole-genome sequenced).

We have added a sentence stating that the genes were well covered, pointing to the table (page 4, line 74):

„(all five genes were well-covered in the probands' sequence data, see Supplementary Table 2)).“

I.b. All variants in these genes should be presented, along with relaxation of MAF filters in these genes given the fact that the population has a founder effect

Answer: In response to the reviewer's comment, we raised the MAF threshold in the six genes to a minor allele frequency of 10%. We find no coding variants in the six genes with a MAF under 10%, in any of the eight p.Tyr2Ter homozygotes.

I.c. How common are EROS homozygous variants in EXAC within and outside Iceland?

Answer: There is no homozygous predicted loss-of-function genotype (nonsense, frameshift, essential splice) in *CYBC1* (EROS) in ExAC or gnomAD. In particular, p.Tyr2Ter is not present in ExAC nor in gnomAD. The relatively high frequency of p.Tyr2Ter (MAF=0.76%) and the subsequently great number of homozygotes in Iceland is probably because of the founder effect.

*In response to the reviewer's comment we have added a supplementary table with all homozygous coding variants we observe in *CYBC1* in our set (Icelandic frequencies and gnomAD frequencies).*

2. There is insufficient clinical and biochemical data provided here for individual patients given that this is intended to describe a new form of CGD.

Answer: We have now expanded on the clinical phenotype of the eight homozygous individuals, as detailed in our response to comment 8 from the reviewer (see below).

3. The authors state that they have the ability to recall patients as part of the gene screening study. In my opinion, all patients should have biochemical confirmation of reduced ROS, and ideally reduced gp91Phox. In addition to consolidating the data shown, biochemical confirmation would be a routine part of making a CGD diagnosis.

Answer: We agree with the reviewer that we would ideally have preferred to have biochemical confirmation of reduced ROS for all eight p.Tyr2Ter homozygous individuals (as well as the 5 *CYBA* homozygotes). However, out of the eight homozygotes, two are deceased, one had bone-marrow transplantation, two were contacted and were unable to come to the phenotyping center, two came (of whom only one was able to come a second time for acquisition of a blood sample for fresh neutrophils).

4.a. It would also be useful to know whether there is preservation of a degree of oxidative burst to explain the somewhat milder infection phenotypes described compared with other genetic causes CGD. For example in Figure 1 there is some evidence of ROS production by the patients- how does this compare with *CYBB* patients?

Answer: We agree that the infectious phenotypes observed in our group of patients could be considered milder than what is typically observed for gp91^{phox} (*CYBB*)-deficient patients. As outlined in our response to a similar comment from Reviewer #1 (comment 1.a.), we believe this might be because of the unusual design of our study of not just genotyping diagnosed CGD patients. Evaluation of all carriers provides a less biased evaluation of penetrance and expressivity than the classical case-only analysis.

However, in response to the reviewer's comment, we have added a text to the Discussion section addressing A) the somewhat milder clinical presentation of the six individuals identified by their genotypes (page 12, line 256) and B) the residual ROS production (page 12, line 245):

A)

„Notably, recruitment based on genotypes can be expected to reveal greater phenotypic variability than recruitment based on clinical presentation alone, as is illustrated by our study.“

B)

„Although the variable clinical spectrum of CGD does not always depend on the gene involved, X-linked CYBB mutations often lead to a more severe ROS defect than autosomal recessive causes of CGD⁵. Our results indicate that CYBC1-deficient individuals have some residual ROS production, similar to what has been reported for other autosomal recessive forms of CGD³⁶. Residual ROS production has been shown to correlate with a milder clinical presentation, and improved overall survival in CGD depending on detection and treatment of the disease³⁶.“

4.b. Is the oxidative burst similarly impaired for all EROS patients or is there variation?

***Answer:** We note that outcome of the DHR assay typically varies somewhat between individual experiments, and having only three tested patients makes it difficult to conclude on this point.*

As part of our response to comment 5.b. from Reviewer #1 we have attempted quantification of the oxidative burst by calculating the stimulation index (SI), calculated by dividing the mean fluorescence of DHR in PMA stimulated cells by unstimulated cells, for the three tested homozygotes.

The stimulation index ranged from 1.34 to 2.50 ($SI_A=1.34$, $SI_B=2.50$, $SI_D=2.35$), which could be considered a similar degree of impairment (all SI lower than 3). Notably, in an earlier study of the correlation between genetic subtypes of CGD and degree of oxidative burst defect (Köker et al., 2013), the mean SI for the more severely affected patients was close to 1, while the less severely affected cases had $SI > 3$.

4.c. How many patients were on immunosuppressive agents at the time that their oxidative burst was done?

***Answer:** None of the three homozygous individuals for whom an oxidative burst test was performed were on immunosuppressive agents at the time of their test.*

5. Figure 3: Why are the homozygous patients not represented on the quantitative graph shown in part a)?

***Answer:** We have now changed Figure 3.a) to include the homozygotes from Figure 3.b).*

6. The authors should explain how there is absent EROS protein in one patient and 50% EROS expression in heterozygous carriers if NMD is not associated with this mutation.

Answer: *The mutation causes a premature termination at the second amino acid of C17orf62, resulting in protein truncation (single amino acid), rather than nonsense-mediated decay of the mRNA. This corresponds to what we would expect to happen, since nonsense-mediated decay has been shown to be inefficient when a premature stop codon is introduced this early (Lindeboom et al., 2016). The absence of the C17orf62 (EROS) protein from p.Tyr2Ter homozygotes, and a 50% reduction in heterozygous carriers, refers to exactly that, i.e. that we did not detect the full-length C17orf62 protein because the product is truncated at the second amino acid.*

7. In supplementary Figure 5 how do the authors explain absence of gp91Phox in Non-carrier D and the discrepancy between neutrophil and macrophage levels of gp91Phox for Homozygote D (Supplemental figure 3)?

Answer: *What appears to be absence of gp91^{phox} from non-carrier D is in fact explained by a weaker band both for gp91^{phox} and ACTIN (the loading control) for this individual. When assessed relative to ACTIN, gp91^{phox} levels in non-carrier D are normal (as shown in Figure 2). This is already stated in the legend of Supplementary Figure 5.*

Although no conclusion can be made with measurements from only one individual (homozygote D), we have now added a speculation on the discrepancy in gp91^{phox} expression levels between neutrophils and macrophages to the Discussion section (page 11, line 225):

„Although no conclusion can be drawn from only one individual, the difference in expression of gp91^{phox} between monocyte-derived macrophages (absent) and neutrophils (50% reduced) from homozygous individual D is noteworthy, and suggests the presence of more than one chaperone for cytochrome b-245 in the neutrophil lineage.“

8. More clinical information should be provided especially for types of infection and severity of bowel disease.

Answer: *We have gathered more information on the severity of the bowel disease for individuals C, E, and F, and have now added the following text to the manuscript (page 6, line 118):*

„Of the three additional homozygotes suffering from colitis, two had a confirmed granulomatous colitis (individuals C and F). Individual C was initially diagnosed with ulcerative colitis (UC), receiving a CD diagnosis only after undergoing total colectomy that revealed extensive granulomatous inflammation of the colon. The third individual, individual E, was diagnosed at 14 years of age with total colitis. Repeated biopsies from colon did not detect granulomatous changes.“

*Additionally, we have received more detailed information on the pathogenic origin of some of the infectious disorders noted among the homozygotes. We have verified that the invasive pneumococcal disease in individual G was caused by *Streptococcus pneumoniae*, now added to the main text (page 7, line 147):*

„Individual G had invasive pneumococcal disease at 30 years of age caused by Streptococcus pneumoniae⁶, and later developed interstitial pulmonary fibrosis, a known manifestation of CGD^{1,15}.“

We also found that the intestinal infection noted for individual B was caused by Clostridium difficile, and have added all known infectious origins to Table 1.

9.a. The infection history shown for the cohort in Table 1 details some organisms and infections that are commonly associated with CGD (eg Burkholderia and disseminated TB) but others that are not, such as viral infections and invasive pneumococcus. There are notable absences for Aspergillus and Staph Aureus infections that are particularly common in other forms of CGD: this should be discussed.

Answer: Again, we would like to emphasize that we are the first study to describe a group of CGD patients identified based on their genotype rather than clinical presentation. Thereby, we do not expect this group of patients to show the same spectrum of infections as those individuals ascertained clinically.

9.b. Additionally some feature are included in the infection column that would be better included under IBD phenotype (eg anal fistula) and in the immunological/inflammatory column that would be better under infection (eg otitis media).

Answer: We thank the reviewer for this suggestion, and have now rearranged Table 1 so that it has a column for gastrointestinal symptoms, where we include all the IBD phenotypes, and have moved the entries „Acute suppurative otitis media“ and „Suppurative otitis media“ to the „Other Infection“ column.

10. The authors should provide a case history and pedigree for each individual in supplementary data to give a better idea of the clinical spectrum associated with the genotype described. Were there other family members sequenced for pedigrees other than the index pedigree?

Answer: In addition to an overall extension of our discussion of the clinical phenotypes observed among the eight p.Tyr2Ter homozygotes, we have now added pedigrees for the six additional homozygous individuals (Supplementary Figure 6) along with an indication of whether or not a parent was whole-genome sequenced and/or chip-genotyped.

11. In general, the text is poorly focused and should be shortened with concentration on flow of the information presented. There is repetition and a relatively small amount of space is taken describing the main findings about EROS defects. For example, much of the text in lines 187 onwards is repetition.

Answer: The manuscript, initially formatted as a letter, has now been modified to fit an Article format with Introduction, Results, and Discussion sections. The manuscript gains from this modified structure, along with the extended word allowance in Nature Communication. In

line with this comment from the reviewer, we have now re-formatted part of our results and discussion, expanding on implications of the *CYBC1* p.Tyr2Ter mutation.

12. In addition the insertions of text about all forms of CGD in Iceland does not particularly add to the manuscript/ data presented here. For example, lines 197-200. It is not clear here whether the 4 previously diagnosed were all *CYBB* mutations or whether the index cases here are included. It is also surprising that patients with *CYBB* mutations were not diagnosed with CGD prior to WGS or chip sequencing as the diagnosis is usually picked up by clinical presentation. I am not sure that this manuscript is the correct place to combine the discussion of CGD in Iceland caused by other known genetic causes as it distracts from the main message.

Answer: We have modified the text (previously in lines 197-200) to clarify which mutation carriers had been previously diagnosed with CGD (page 12, line 252):

*„Out of the thirteen Icelanders with a CGD genotype, only the two probands and two of the *CYBA* p.Arg90Trp homozygotes were, to our knowledge, diagnosed with CGD prior to the discovery of *CYBC1*“*

Since we have genotypic information on such a large percentage of the Icelandic population (155K chip-genotyped, of which 37K have been WGS) we can identify carriers of disease-causing genotypes. These individuals have submitted their sample as part of the deCODE sequencing initiative in Iceland, not through clinical sequencing.

*We feel that previous comments from the reviewer (comments 1.b.-1.c.), on the importance of assessing genotypes in other CGD genes, are contradictory to his suggestion that the *CYBA* p.Arg90Trp founder mutation should not be discussed in this manuscript.*

13. Information about the Saudi-Arabian patient is interesting but HLH is usually the consequence of infection in CGD. Can the authors be contacted to clarify the clinical phenotype of this patient- for example did the patient have colitis?

Answer: A more detailed clinical description was provided for the Saudi-Arabian patient in supplementary material of that publication; a one year old male with growth retardation / short stature, lymphadenopathy, hepatosplenomegaly, recurrent fever, anemia/neutropenia/pancytopenia. The patient had a suspicion of hemophagocytic lymphohistiocytosis (HLH), but the diagnosis was ruled out. There was not mention of any gastrointestinal symptoms for that patient. We have now changed the sentence of HLH preceding CGD to the following (page 10, line 213):

„Interestingly, the individual homozygous for that mutation had symptoms compatible with CGD, including lymphadenopathy, hepatosplenomegaly, recurrent fever, and growth retardation³⁰“

Minor comments:

14. The word ‘imputation’ is unclear in meaning for a broad audience and should be re-phrased/explained. Similarly, the phrase ‘(imputation information = 0.99)’ should be explained for a non-genetic audience- or removed if not relevant.

Answer: We have now added an explanation of imputation, and imputation information, to the Methods section:

“Imputation is a method for inferring genotypic status of variants that have not been directly genotyped. Imputation of untyped variants into the mix of typed variants is now routine in human genetics⁷. The extensive genealogical information available for the Icelandic population, along with deCODE’s long-range phasing of a large set of genotyped Icelanders^{7,41} has increased imputation accuracy and speed by removing uncertainty in phasing. In general, a haplotype can be imputed into an untyped individual if two genotyped relatives share a long haplotype that is identical by descent (IBD), and the genealogy indicates that the path of IBD sharing goes through the individual⁴¹.

The informativeness of genotype imputation is estimated by the ratio of the variance of imputed expected allele counts and the variance of the actual allele counts:

$$(Var(E(\theta | \text{chip data}))) / (Var(\theta)),$$

where θ is the haplotype allele count. $Var(E(\theta | \text{chip data}))$ was estimated by the observed variance of the allele imputations and $Var(\theta)$ was estimated by $p(1-p)$, where p is the allele frequency. This metric has the property of being 0 when the imputation is completely uninformative (the same value is always imputed) and 1 when the imputation is fully informative. Asymptotically, the expectation of the metric equals the r^2 between the imputed alleles and the true genotypes.”

15. In the abstract, it is not clear whether the following sentence refers to analysis within the EROS mutation cohort or across the whole genome/gene chip data sets:

‘Homozygosity for p.Tyr2Ter associates with inflammatory bowel disease (IBD) (P = 8.3 × 10⁻⁸; OR (95% CI) = 40 67.6 (14.5, 315.5)), and growth retardation (P = 3.3 × 10⁻⁴; Effect (95% CI) = -1.24 SD (-41 1.92, -0.56) / -8.5 cm), consistent with mutations in other known CGD genes’

Answer: The eight CYBC1 p.Tyr2Ter homozygotes are part of the chip genotyping set (155K chip-genotyped, of whom 37K have also been WGS). The association tests performed were based on the whole set, including the eight homozygotes, and are merely a way of assigning a numerical value to the excess of diseases / deviation of traits associated with this genotype.

To make it more clear that the association tests are not independent of the group of homozygotes we have added „Consequently“ to the beginning of the sentence (page 2, line 34):

„Consequently, homozygosity for p.Tyr2Ter associates with inflammatory bowel disease (IBD) in Iceland ...“

16. Lines 71-76 would be better combined with the text prior to description of the index cases (lines 49-59)

Answer: As suggested by the reviewer, we have now moved lines 71-76 to what now corresponds to lines 41-46.

17. Line 93: GCGR should be explained

Answer: We have added a short explanation on why the GCGR genotype was dismissed (page 5, line 89):

“One of these, a homozygous missense mutation in GCGR encoding the human glucagon receptor (NM_000160.3:c.449G>A; NP_000151.1:p.Ser150Asn; hg38 position chr17:81,811,277; MAF=0.33%), was dismissed based on lack of biological relevance (Supplementary Information).”

We also discuss GCGR in more detail in the Supplementary Material, under “Variants detected in the two probands”:

„The missense variant in GCGR, NP_000151.1:p.Ser150Asn, is GR homozygous in three other Icelanders. GCGR encodes a glucagon receptor, involved in the regulation of blood glucose levels and glucose homeostasis. A single missense variant in GCGR, NP_000151.1:p.Gly40Ser, has been reported to associate with type 2 diabetes. No other phenotypic traits have been linked to homozygosity of variants in GCGR in humans. Homozygous knockout mice display abnormal glucagon levels with other phenotypes secondary to abnormal glucagon levels².“

18. Line 103/4: the final sentence here is superfluous.

Answer: The HGNC has now approved the gene symbol CYBC1 for the former placeholder C17orf62. We have now removed the sentence mentioned by the reviewer and added the HGNC accession number to the Data availability statement in the Methods section:

„The HGNC has accepted the gene symbol CYBC1 instead of the placeholder C17orf62, under the accession number HGNC:28672.“

URL: https://www.genenames.org/cgi-bin/gene_symbol_report?hgnc_id=HGNC:28672

19. Lines 128/129: ‘In patients with CGD, 129 microbial catalase is traditionally recognized as the most important virulence factor¹⁴’ should be clarified ‘for infection’.

Answer: We have modified the sentence as suggested by the reviewer:

„In patients with CGD, microbial catalase is traditionally recognized as the most important virulence factor in infections¹⁶.“

20. Lines 133-137: This sentence could be removed.

Answer: Overall, the four reviewers are in favour of providing maximum phenotypic information in the manuscript. We do consider that the frequency of the listed infections in Iceland is part of this phenotypic description. We are using the information on the overall population as a phenotypic normative set.

21. Line 138: How confident are the authors of the clinical phenotype provided for patient G: pulmonary fibrosis is seen in CGD, often as a result of previous infections- did this patient have a history of lung infections? Also pneumococcal disease is not one of the ‘common/typical’ CGD infections.

Answer: Individual G likely suffered from pneumonia, as a part of her invasive pneumococcal disease at age 30 (in 1985). However, although our infectious registry dates back to 1985, our discharge diagnoses do not. Since we do not have positive cultures from sputum (only from blood and cerebrospinal fluid) we cannot verify that individual G had pneumonia.

*We have, however, added information on the pathogenic origin of some of the infectious episodes to Table 1. This includes the origin of the invasive pneumococcal disease in individual G which was caused by *Streptococcus pneumoniae* (identified as a positive blood culture). *S. Pneumoniae* has been described to cause infection in CGD (van den Berg et al., 2009).*

As noted by the reviewer, pulmonary fibrosis is seen in CGD since the exaggerated inflammatory responses of CGD may contribute to scar formation and fibrosis in the lung (Mahdavian et al., 2013), and we note that our health information for this individual is incomplete, and a history of lung infections can therefore neither be confirmed nor ruled out.

22. Lines 191-193: is this a fair way to estimate prevalence as presumably CGD patients diagnosed by candidate gene sequencing following recognition of the classical clinical phenotype would not then be submitted for WGS/gene chip: could the selection of patients for genetic studies specifically exclude those with known diagnoses?

Answer: Although we have genotypic information on approximately half of the Icelandic population, gathered more or less at random, we can of course not exclude that we are missing some cases. We have added “predicted” to the sentence, which now reads (page 11, line 239):

*“Overall, we observe thirteen Icelanders with a homozygous genotype causing CGD, eight homozygous for *CYBC1* p.Tyr2Ter and five for *CYBA* p.Arg90Trp, amounting to a predicted CGD prevalence of 1 in 12,000 in Iceland, much higher than reported in the literature¹.”*

We note that we were already stating that this prevalence is higher than reported elsewhere.

23. Line 197: sentence starting ‘In particular..’: I don’t understand this as there are already 5 CYBB patients alluded to in the preceding text.

Answer: We have now modified this sentence as detailed in our response to comment 12 from the reviewer.

24. Figure 3: it would be more intuitive to show homozygote results before heterozygote results.

Answer: We have now modified Figure 3 so that the first subplot includes the expression of CYBC1 in lymphocytes from homozygous individuals D and F, as well as from heterozygotes.

25. Supplementary Table 1: I do not think this adds to this paper and I would remove it along with discussion of CYBB patients in the text.

Answer: As already detailed in our response to comment 12 from the reviewer, we believe Supplementary Table 1 is compliant with the reviewer's previous suggestion on showing the genetic diversity in other CGD genes in Iceland (comments 1.b.-1.c.).

Reviewer #3 (Remarks to the Author):

This manuscript sheds further light into the pathogenesis of CGD, expanding the list of genes implicated in the disease and providing clinical data to support findings in mouse models that have uncovered a new mechanism for CGD.

1. Overall, the impact of the findings could be strengthened with additional functional data on the neutrophils of these homozygous donors, by evaluating microbial killing, NET formation and hyperinflammation (ROS suppresses cytokine production in response to small microbes). These data would help solidify the functional association between the defects in ROS burst and the clinical pathology (recurrent infections, hyperinflammation) reported for these individuals.

Answer: We thank the reviewer for these comments. Unfortunately, neutrophils were only available to us from one homozygous donor. Out of the eight homozygotes two are deceased, one had bone-marrow transplantation, three were contacted and were unable to come to the phenotyping center, two came to the center for phenotypic assessment, and only one was able to come a second time for acquisition of a blood sample (allowing for the harvesting of fresh neutrophils).

2. Pg. 3 ln 70. It would be useful to know how long the BM transplant recipient has been symptom free.

Answer: According to the pediatrician overseeing the two probands, the BM transplant recipient has been symptom-free (as of May 2018) since his transplatation (in August 2010). We have now clarified this in the main text as follows (page 4, line 68):

„The older brother (individual B) died of post-HSCT complications, whereas the younger brother (individual A) was successfully transplanted in 2010 and has been symptom-free since then.“

We have also modified the detailed clinical description, presented in the Supplementary Material, accordingly.

Reviewer #4 (Remarks to the Author):

The manuscript by Arnadotti et al. reports a new disease-causing gene for CGD, EROS, which was identified in 8 patients from Iceland diagnosed with CGD. These patients were homozygous for a loss-of-function mutation, p.Tyr2Ter, which produces truncated protein. This variant is relatively common (MAF= 0.76%) in the Iceland population and absent in the gnomAD database. Heterozygosity for this variant does not cause a decrease in the EROS mRNA expression, although the protein expression appears to be reduced by 50%. Individuals homozygous for this variant have null protein expression. Together, these data provide solid evidence that this variant is pathogenic.

In addition to this variant being linked to CGD, it is strongly associated with IBD under the recessive model of inheritance with the impressive odds ratio OR=67.6. Homozygosity for this mutation is also associated with reduced height in the same population, which could be reflection of the inflammatory phenotype in these individuals.

The authors showed absence of mutant EROS protein in lymphocytes of two patients, decreased oxidative burst in neutrophils of 3 patients including the asymptomatic individual D. Protein expression of the known CGD gene that is regulated by EROS, gp91 phox, was reduced in primary cells of homozygous individuals (one affected and one unaffected). These functional experiments further support the genetic evidence that LOF mutations in EROS are linked to CGD and/or IBD. Mice deficient for Eros are susceptible to inflammation which is consistent with the infectious phenotype in patients with CGD.

In summary, this is a fairly straight forward genetic study that has benefited from the exquisite data collection (genetic and medical records) in the Iceland population.

Comments

Genetics:

1. Are there other genes in the index family that passed the filtering criteria beside the two reported genes, GCGR and EROS? If so, please show them in the supplementary table. As the filtering at allele freq. <2% is not very stringent, although it is acceptable for studies in founder populations, it is possible that these two patients may have pathogenic mutations in other genes that are linked to immunodeficiency (typically recessively inherited disorders and more common in founder populations).

Answer: When selecting variants for analysis we filtered on a minor allele frequency of 0.05% for heterozygous genotypes, and excluded heterozygous genotypes present in either of the unaffected parents, and filtered on a minor allele frequency of 2% for homozygous/compound heterozygous genotypes. We found no other variants meeting these filtering criteria than the two mentioned in the main text, i.e. homozygous missense variants in GCGR and C17orf62/CYBC1 (EROS).

In response to the reviewer's concerns, we performed an additional analysis on known CVID genes (25 genes outlined by Bogaert et al., 2015) in the two probands, relaxing the MAF threshold to 4%. We found no variant in these 25 genes in the two probands with MAF below 4%.

2. Please discuss why the individual D have no disease features despite being homozygous for the variant and having abnormal neutrophil oxidative burst test. Are there for example variants in her exome that may function as modifiers or protective variants for IBD? Are there protective variants for IBD from GWAS studies?

Answer: Firstly, we note that gastrointestinal symptoms (often mimicking Crohn's disease) are only seen in approximately 40% of individuals with CGD. Thus, it is not surprising that individual D does not have such symptoms.

Secondly, that individual D has not yet reported any severe infections (as we already note in the manuscript) can be explained by A) she is among the youngest homozygotes (apart from the two probands) and B) our records are not complete, and mainly based on hospital discharge diagnoses (also noted in the manuscript).

Lastly, and most importantly, we believe her lack of gastrointestinal symptoms and severe infections can be partly explained by the unique design of our study. Most of the patients in our cohort (6/8) were identified on the basis of their genotype, in contrast to earlier studies involving the genetics of CGD, where individuals were recruited based on their clinical presentation. We consider it likely that individuals exist who carry a known pathogenic CGD genotype (mutations in CYBA, CYBB, NCF1, NCF2, or NCF4) without being diagnosed with the disease or showing extreme phenotypes, similar to individual D.

3.a. The allele frequency for the Tyr2Ter mutation is almost 1% in this population (q2 is 1:10,000) and 8 homozygous patients with likely CGD phenotype were identified in a cohort of about 180.000 people. However, the total number of homozygous individuals in the entire data set of ~340,000 people is unclear. Homozygosity for this variant is strongly associated with IBD and reduced height, which suggests that there could be more homozygous individuals in the studied population.

Answer: We have added the following sentence to clarify the number of p.Tyr2Ter homozygotes in our set (155K chip-genotyped and imputed, of whom 37K have also been WGS), and the expected number of homozygous individuals in the whole of Iceland (338K individuals) (page 11, line 237):

„We have identified eight Icelanders homozygous for the p.Tyr2Ter mutation in our dataset, comprising 155K chip-genotyped individuals, and would expect 19 homozygotes in the entire Icelandic population (338K individuals).”

3.b. The association data for IBD and reduced height should be shown in table format i.e. allele frequencies in patient and control cohorts. Please clarify whether the 8 patients with CGD are also included in these association studies.

Answer: We have now added a table (Supplementary Table 4) showing the association results, where we include the p.Tyr2Ter homozygous genotype frequencies within and outside A) the IBD cohort and B) the cohort of short individuals (defined by us as the bottom 15%). In addition, we have added a table (Supplementary Table 5) that shows the standard deviation of each homozygous individual from the average of 96,029 Icelanders with available height measurements.

To clarify, yes the eight p.Tyr2Ter homozygotes are included in the association study as they are part of our chip-genotyping set (see also our response to comment 15 from Reviewer #2).

3.c. Most GWAS-identified variants for IBD have OR in the range of 1-3, which raises the question whether the EROS gene is actually a new IBD recessively inherited gene. In support of this hypothesis is the fact that 5/7 symptomatic patients diagnosed with CGD have IBD and that infectious history is not very strong in all 8 patients (patients C, E, F have none or viral infections) (table 1). Patients A and B have strong clinical history and functional evidence (burst test) for CGD while other patients are less convincing. This is understandable as their clinical data was not complete and is based on self-reported medical records. Please comment.

Answer: We conclude that homozygosity for loss-of-function mutations in CYBC1 causes CGD, and we and others conclude that close to half of CGD patients also show phenotypes resembling IBD. We note that atypical/severe infections are seen in at least five patients in our cohort, e.g. legionnaires' disease, candidal septicaemia, invasive pneumococcal disease at 30 years of age, and military tuberculosis, in addition to the severe intestinal infections noted in Table 1. This, together with the biochemically confirmed oxidative burst defect, suggests that the genetic contribution is not limited to IBD.

In response to the reviewer's comment we have added the following sentence to the main text (page 7, line 149):

„Thus, in addition to the frequency of gastrointestinal symptoms among the eight homozygous individuals, an excess of atypical and severe infections was evident.“

Additionally, we have added a speculative comment to the Discussion section on whether the gastrointestinal symptoms might be more pronounced in CYBC1-deficient patients (page 11, line 226):

„Although no conclusion can be drawn from only one individual, the difference in expression of gp91^{phox} between monocyte-derived macrophages (absent) and neutrophils (50% reduced) from homozygous individual D is noteworthy, and suggests the presence of more than one chaperone for cytochrome b-245 in the neutrophil lineage. Consequently, mechanisms more specific to macrophages, such as the exaggerated proinflammatory responses seen in CGD patients³³, might become a more prominent phenotype in CYBC1-deficient individuals.“

4. Functional studies:

Although several experiments showed the effect of p.Tyr2Ter on the protein expression and function, this manuscript is not very strong with functional data.

4.a. The oxidative burst experiments are fairly convincing. There is some ambiguity regarding the mRNA expression and protein expression of mutant EROS. NMD bypass of protein truncating mutations that are close to the start codon may explain normal mRNA expression in carriers for p.Tyr2Ter (Lindeboom et al, 2017). Despite no difference in the mRNA or allelic expression of mutant EROS in the heterozygous carriers vs non-carriers, the protein expression is decreased in mutation carriers based on the quantification of western blots, which are otherwise not convincing (supplementary figure 4). Could there be an alternatively translated EROS protein as the result of downstream re-initiation in translation? Does anti-C17orf62Ab detect the N-terminal or the C- terminal end of the protein?

Answer: In response to the reviewers comment on ambiguity regarding the mRNA expression and protein expression of mutant CYBC1 (EROS), along with the speculation of an alternatively translated EROS protein, we have now extended our discussion on the effect on the protein as follows (page 8, line 165):

“This indicates that p.Tyr2Ter does not lead to nonsense-mediated mRNA decay (NMD), consistent with NMD being less efficient when premature termination codons occur in such close proximity to the start codon²⁵. Rather, we expect p.Tyr2Ter to result in a truncated product at the protein level, consisting of a single amino acid. We investigated the possibility of an alternative start codon in C17orf62 producing a functional protein. The closest codon for methionine following the same reading frame as p.Tyr2Ter occurs in exon 6 (out of 7 exons). The non-coding mRNA corresponding to this start codon (NR_036518, expressed in whole blood²³) encodes a sequence of 63 amino acids, that would lack the two membrane-spanning domains of C17orf62.”

The anti-C17orf62Ab recognizes amino acids 125-187, i.e. the C-terminal end of the protein. If the potential alternative initiation codon discussed above would produce a protein, the antibody would recognize it and we would have seen a band of ~7kDa on the western blot (a rough estimate for a 63 amino acid peptide chain), but we do not.

4.b. What is the mRNA expression in homozygous individual F or D? (their primary cells were available for other experiments) Does mRNA expression in homozygous individuals correlate with the absent mutant protein?

Answer: Homozygous individuals D and F show now signs of reduced mRNA expression. This is not surprising since, as noted by the reviewer in comment 4.a., premature termination codons in such close proximity to the start codon are not expected to result in NMD, but rather protein truncation.

However, we have already shown that the termination at the second amino acid results in absence of the full-length CYBC1 protein, so the mRNA expression levels for the two homozygotes are irrelevant.

4.c. What is the explanation for inconsistency in Gp91phox protein expression in the primary cells of individual D as shown in sup. fig 3. and 5 (50% reduction vs. null expression). Both experiments were done in myeloid cells that highly express EROS and

Gp91phox, thus this discrepancy is unlikely due to cell-type specific expression. It also appears that the same Gp91phox antibody was used for both western blots.

Answer: In response to this comment from the reviewer we have added a speculative thought to the Discussion section (overlapping with our response to comment 3.c. from the reviewer). Since we could only measure gp91^{phox} expression in neutrophils from individual D, no concrete conclusion could be drawn (page 11, line 226):

„Although no conclusion can be drawn from only one individual, the difference in expression of gp91^{phox} between monocyte-derived macrophages (absent) and neutrophils (50% reduced) from homozygous individual D is noteworthy, and suggests the presence of more than one chaperone for cytochrome b-245 in the neutrophil lineage.“

4.d. This study is a bit short of details on the function of human EROS protein, which is still largely unknown. The authors discussed and speculated its function based on the murine model of Eros deficiency. Primary patients cells are available to replicate some experiments from the murine model study. What is the mechanism by which EROS regulates the expression of gp91phox and gp22phox? Studies in KO mice suggested that Eros might regulate the proteosomal degradation of gp91 phox and gp22phox. Is that the case in human cells? Readers of this journal would appreciate more discussion and data on the EROS function in humans.

Answer: Due to the extended word allowance offered by Nature Communications we have now expanded our discussion on the likely role of CYBC1 (EROS) as a chaperone protein. The modified text now reads as follows (overlapping with comments 3.c. and 4.c. from the reviewer) (page 11, line 221):

„A drastic reduction in gp91^{phox} expression levels in p.Tyr2Ter homozygous individuals, the oxidative burst defect, and prior evidence in knockout mice all point to CYBC1 as an essential factor for successful formation of the NADPH oxidase complex. CYBC1 is known to co-localize and interact with gp91^{phox} in the endoplasmic reticulum (ER)^{2,32}, likely acting as a chaperone for dimerization of gp91^{phox} and p22^{phox} (cytochrome b-245). Although no conclusion can be drawn from only one individual, the difference in expression of gp91^{phox} between monocyte-derived macrophages (absent) and neutrophils (50% reduced) from homozygous individual D is noteworthy, and suggests the presence of more than one chaperone for cytochrome b-245 in the neutrophil lineage. Consequently, mechanisms more specific to macrophages, such as the exaggerated proinflammatory responses seen in CGD patients³³, might become a more prominent phenotype in CYBC1-deficient individuals.“

5. Other suggestions:

5.a. Figure 1: please add the panel showing EROS protein structure with domains and the location of the causal variant.

5.b. Figure 2: not informative and it should be moved to supplementary. It could be replaced with the table showing data from association studies.

Answer: *We have now moved Figure 2 to be part of the panel shown in Figure 1. This shows a topological prediction of the CYBC1 (EROS) protein, and the two known domains, i.e. the transmembrane domains. It also shows the location of the causal variant (p.Tyr2Ter). No crystal structure is available for this newly characterized protein.*

We have added the table showing data from association studies to the Supplementary (Supplementary Table 4).

Reviewer #1 (Remarks to the Author):

To the authors:

The paper is improved but needs further revision to better depict the cellular and clinical features of the homozygous CYBC1 mutant patients vs patients with autosomal or X-linked recessive null mutations in CYBB, CYBA, NCF1, and NCF2, who have a distinctive disease phenotype referred to as chronic granulomatous disease.

General comment: the authors seem to want to make the story fit into a “CGD box” and oversimplify clinical features and what appears to be a differential impact of CYB1C loss on gp91phox. From the data presented, it appears that the CYBC1 patients have a distinct syndrome, resembling an atypical form of CGD with little predilection to microbes associated with “classic” CGD but still with some inflammatory manifestations, particularly GI. These nuances reveal important features of human biology and the role of the NADPH oxidase in responses to microbes and inflammation and should not be oversimplified.

As noted in my prior review, “classic” CGD is characterized by recurrent severe bacterial and fungal infections, often life-threatening, involving skin/soft tissue and their draining lymph nodes, lung, and abscesses in the liver, bone, or brain. A distinctive spectrum of microbes cause infections in CGD. This distinctive spectrum indicates that only certain microbes have non-redundant requirement for ROS for their control. This is not entirely explained by catalase activity. In addition, CGD patients frequently have inflammatory manifestations, including inflammation in the GI tract that resembles Crohn’s disease. This is believed to reflect the influence of NADPH oxidase ROS on regulating inflammatory signaling and other processes not directly related to their microbicidal effect.

This phenotype is reported in ample literature involving thousands of patients and from many different regions of the world.

Almost all CGD patients have entirely absent oxidase activity or, in the case of NCF1 null alleles, profoundly reduced oxidase activity, in all leukocytes expressing the NADPH oxidase. The NCF1 protein is an adaptor protein rather than being part of the catalytic machinery as for CYBB, CYBA, and NCF2, which is believed to explain why small amounts of oxidase ROS are detectable. However, NCF1 patients still get recurrent, severe invasive deep tissue or lung infections with the same distinctive group of microbes as seen in the other genetic forms; just tend to occur at a lower frequency, and the infections are often not quite as severe, explaining lower mortality of this genetic subgroup. Hypomorphic CYBB mutations associated with residual oxidase activity have also been described, and are often less clinically severe. There is some variation even with full null /oxidase-negative patients – eg first “CGD” infection at 6 years of age vs 6 months of age, but again, the sites and organisms involved are distinctive.

Only very rarely has a diagnosis in adult life been made, typically with the occurrence of an infectious or inflammatory manifestation of CGD (often these are NCF1 -null patients or occasionally hypomorphic mutations in CYBB) . These scenarios are unusual enough to be published as case reports; in many instances, there were prior infections characteristic of CGD (eg Nocardia, Klebsiella,) and the diagnosis CGD was overlooked.

The fifth group of NADPH oxidase defects are patients with mutations involving the specialized regulatory subunit NCF4. These patients have an atypical presentation, likely due to incomplete loss of oxidase activity. The first patient (Matute, BLOOD, presented with granulomatous GI disease, the patient did not have any serious deep-tissue or lung infections at time of diagnosis at age 3, and it was recognized as an atypical form of CGD -- as no prior patients with NCF4 deficiency and typical CGD infections had mutations involving NCF4. A recently published larger series has borne this out (Van de Geer et al JCI 2018). This study concluded that. "...the patients suffer from hyper-inflammation and peripheral infections, but they do not display any of the invasive. bacterial and fungal infections seen in CGD. Inherited p40phox deficiency underlies a distinctive condition, resembling a mild, atypical form of CGD."

Other comments:

--- Abstract and throughout the manuscript:

The study will be better served by describing the results more precisely --- for example, in the Abstract: "... drastic reduction in the NADPH's main subunit, gp91phox" -- is not entirely true -- the single PMN sample with Western blot done shows substantial expression of gp91phox, where as in two patients, no gp91phox was detected in monocyte derived macs. -- e.g. stating "impaired expression" may be better. (Also, should be "NADPH oxidase's main subunit" not NADPH's).

-----I disagree with the authors' statement... "... that both phenotypic variability and variable expressivity are accepted for the other known CGD genes." Distinctive features are in fact recognized as a related but different condition. E.g. see above re NCF4 genetic subgroup.

---The author's statement in the Introduction, " The presentation of CGD is highly variable, ranging from relatively mild infections in adulthood to fatal septicemia in infancy"... is not an accurate assessment of the literature.

-----In the CYBC1 series presented in this manuscript, the clinical phenotype of the majority of patients does not resemble classic CGD from the infectious disease standpoint -- e.g the nature and severity of infection, distinctive spectrum of microbes. The CYBC1 patients are described as having "an excess of atypical and severe infections" -- perhaps these are atypical for a healthy person, but most of these are not characteristic of CGD.

The index pair of brothers (apart from their significant GI disease,) had some infectious manifestations seen in "classic" CGD: documented candida septicemia, and episode of lymphadenitis

and subcutaneous tissue infections, although no pneumonia or other deep tissue infections. The Burkholderia was cultured from a mouth ulcer, notably, in CGD, Burkholderia infections in CGD are typically are invasive lung and soft tissue infections. Legionnaire's disease, described in one brother, is only rarely described in CGD, so may be a coincidence.

As far as other patients:

- One had miliary TB, a pathogen that CGD patients have increased susceptibility to.
- Another patient had invasive *S. pneumoniae* as manifested by a positive blood culture. However, *S. pneumoniae*, while described in a few patients with CGD, is not an organism that CGD patients have increased susceptibility to. And, as a group, patients with pneumococcal sepsis are highly unlikely to have CGD.
- Increased viral infections are not associated with CGD.
- Otitis media, chronic acne or repeated onychomycosis are not deep tissue/ invasive infections.

-----That numerous cases with null alleles of *CYBB*, *CYBA*, *NCF1* and *NCF2* have been missed due to an ascertainment bias seems exceedingly unlikely. Conversely, if homozygous *CYBC1* could manifest as a classic CGD phenotype, it seems surprising that none in these series had deep tissue or lung infections due to *Staph*, *Aspergillus*, or *B. cepacia*.

-----A more likely explanation for the discovery of patients with a genetic defect involving the NADPH oxidase through "recruitment based on genotypes" is that loss of *CYBC* results in a related but different phenotype because there is residual NADPH oxidase activity, particularly in PMNs, that is protective from infections with microbes characteristic of CGD.

This cellular phenotype (which unfortunately is based on relatively limited available data) is consistent with studies in the EROS KO mouse, which was identified originally by its susceptibility to a macrophage pathogen, *Salmonella*, where intact macrophage function is particularly important. The only other organism the EROS mice were challenged with was *Listeria*, and again, macrophages play an important role in control. It's a bit misleading to state that the Eros KO mice show a phenotype similar to, e.g., *CYBB* KO mice, with a "high susceptibility to infections". Only two organisms were studied, and not those where some oxidase activity in PMNs appears crucial e.g. *Aspergillus*.

Interestingly, one patient in the current series had miliary TB,; macrophages are very important for control of *Mtb*, and infections restricted to mycobacteria were described in several kindreds with

point mutations in CYBB resulting in profoundly impaired gp91phox/ p22phox expression in macrophages but not neutrophils (Bustamante et al Nature Immunol 2011).

---Better acknowledgement in the results/ discussions of the limitations of the cellular phenotype characterization in the current study - very limited analysis of oxidase activity both in number of patients studied and assays performed – only neutrophil DHR assays in two kindreds and to a non-physiologic stimulus, PMA (currently only in the methods). The latter is important as in the EROS mouse, PMN oxidase activity in response to the fungal particle zymosan was only ≈50% reduced. Western blot data on gp91phox expression is also very limited, as previously noted.

---p. 9 not accurate to say that Individual D is “positive for CGD”...first of all, CGD is a clinical phenotype.

--. 10 -only autosomal recessive NCF1 tends to be milder (but with infections involving organisms and sites just like the NCF2, CYBB, and CYBA groups, in contrast to most of the CYBC1 patients); AR NCF2 and CYBA resemble XR CYBB.

--Supplement p. 1. Fig 1b is not an intracellular killing assay

Reviewer #2 (Remarks to the Author):

The revised manuscript is much improved and I am satisfied that my comments have been adequately addressed.

I note that the supplementary figures still refer to EROS rather than CYBC1.

Reviewer #4 (Remarks to the Author):

This study describes a novel homozygous truncating mutation in the CYBC1 gene in 8 patients with clinical features consistent with a CGD phenotype. The p.Tyr2Ter variant is relatively common in the

Icelandic population due to a founder effect. The first two affected siblings were recruited for the study based on their severe clinical phenotype, while the other 6 patients were identified based on their imputed genotype. Their clinical presentation is more variable and ranges from the asymptomatic patient D to two patients with a biopsy proven granulomatous colitis and two patients with recurrent infections not typical for CGD. Detailed clinical records were not available for all patients due to the retrospective data collection. No other candidate pathogenic variants were identified in the index family. None of these 8 patients carry pathogenic variants in other 5 CGD-associated genes or PID-linked genes. The genetic evidence that CYBC1 is a novel gene for CGD is fairly strong. Previous mouse model studies and this study suggest that CYBC1 functions as a chaperone for the cytochrome b-245 complex.

Functional data are somewhat limited due to the lack of primary patients' cells. Experiments were not consistently performed in all cell lineages. Neutrophil oxidative burst defect was confirmed in 3/8 patients, including the asymptomatic patient D.

The RNA and protein expression of CYBC1 were consistent with the loss of function mutation identified in these patients. The Authors concluded that CYBC1 may function as a macrophage-specific chaperone for the formation of the NADPH oxidase complex.

This finding will be of great interest to the research community and clinicians. Ultimately, it may help identify other patients with CGD in countries that do not have abilities to do sophisticated population genetic studies.

Reviewers' comments:

Reviewer #1 (Remarks to the Author):

To the authors:

1. The paper is improved but needs further revision to better depict the cellular and clinical features of the homozygous CYBC1 mutant patients vs patients with autosomal or X-linked recessive null mutations in CYBB, CYBA, NCF1, and NCF2, who have a distinctive disease phenotype referred to as chronic granulomatous disease.

Answer: We want to thank the reviewer for constructive comments that we feel have significantly improved our manuscript. Throughout the revised manuscript, we have attempted to better clarify the phenotypic entity associated with CYBC1 deficiency in comparison to existing literature.

In line with this, we have also made slight modifications to the title of our manuscript:

„A homozygous loss-of-function mutation leading to CYBC1 deficiency causes chronic granulomatous disease“

2. General comment: the authors seem to want to make the story fit into a “CGD box” and oversimplify clinical features and what appears to be a differential impact of CYB1C loss on gp91phox. From the data presented, it appears that the CYBC1 patients have a distinct syndrome, resembling an atypical form of CGD with little predilection to microbes associated with “classic “ CGD but still with some inflammatory manifestations, particularly GI. These nuances reveal important features of human biology and the role of the NADPH oxidase in responses to microbes and inflammation and should not be oversimplified.

Answer: In response to the reviewer's comments, in the revised manuscript we now describe the disease associated with CYBC1 deficiency to manifest mainly with gastrointestinal inflammation and a distinct infectious profile. This is summarized in the final sentence of the Abstract (Abstract, page 2, line 40):

„CYBC1 deficiency results in CGD with a clinical presentation characteristic of macrophage dysfunction, with gastrointestinal inflammation and a distinct profile of infections.“

3. As noted in my prior review, “classic” CGD is characterized by recurrent severe bacterial and fungal infections, often life-threatening, involving skin/soft tissue and their draining lymph nodes, lung, and abscesses in the liver, bone, or brain. A distinctive spectrum of microbes cause infections in CGD. This distinctive spectrum indicates that only certain microbes have non-redundant requirement for ROS for their control. This is not entirely explained by catalase activity. In addition, CGD patients frequently have inflammatory manifestations, including inflammation in the GI tract that resembles Crohn's disease. This is believed to reflect the influence of NADPH oxidase ROS on regulating inflammatory signaling and other processes not directly related to their microbicidal effect.

Answer: In the revised manuscript we have expanded on the description of an unusual microbe spectrum, observed among the 8 CYBC1 p.Tyr2Ter homozygous individuals. See our detailed answers to comments 10.a, 12.a, and 12.b below from this reviewer.

4. This phenotype is reported in ample literature involving thousands of patients and from many different regions of the world.

Almost all CGD patients have entirely absent oxidase activity or, in the case of NCF1 null alleles, profoundly reduced oxidase activity, in all leukocytes expressing the NADPH oxidase. The NCF1 protein is an adaptor protein rather than being part of the catalytic machinery as for CYBB, CYBA, and NCF, which is believed to explain why small amounts of oxidase ROS are detectable. However, NCF1 patients still get recurrent, severe invasive deep tissue or lung infections with the same distinctive group of microbes as seen in the other genetic forms; just tend occur at a lower frequency, and the infections are often not quite as severe, explaining lower mortality of this genetic subgroup. Hypomorphic CYBB mutations associated with residual oxidase activity have also been described, and are often less clinically severe. There is some variation even with full null /oxidase-negative patients – eg first “CGD” infection at 6 years of age vs 6 months of age, but again, the sites and organisms involved are distinctive.

Answer: In the revised manuscript we suggest that the phenotypes of our homozygous individuals may result from macrophage-specific defects. This suggestion is based on the complete lack of expression of gp91^{phox} in macrophages from two p.Tyr2Ter homozygotes, but only a 50% reduction in neutrophils from one of these individuals, as well as the fact that the infection profile of the homozygotes seems to be more consistent with a macrophage defect. See our detailed answers to comment 12.b from this reviewer.

We do still note the severe deficit in ROS production in neutrophils from the 3 tested homozygotes, see our detailed answer to comment 13 from this reviewer.

5. Only very rarely has a diagnosis in adult life been made, typically with the occurrence of an infectious or inflammatory manifestation of CGD (often these are NCF1 -null patients or occasionally hypomorphic mutations in CYBB) . These scenarios are unusual enough to be published as case reports; in many instances, there were prior infections characteristic of CGD (eg Nocardia, Klebsiella,) and the diagnosis CGD was overlooked.

Answer: In the revised manuscript we address the fact that the mean age at onset of symptoms is higher than what is observed for X-linked CYBB deficient CGD patients (Discussion, page 14, line 295):

“The mean age at onset of symptoms for the five p.Tyr2Ter homozygous individuals who had gastrointestinal manifestations is 12.2 years, and deviates somewhat from X-linked recessive CGD patients, the vast majority of whom are diagnosed before 5 years of age².“

See also our answer to comment 11.a from this reviewer.

6. The fifth group of NADPH oxidase defects are patients with mutations involving the specialized regulatory subunit NCF4. These patients have an atypical presentation, likely due to incomplete loss of oxidase activity. The first patient (Matute, BLOOD, presented with granulomatous GI disease, the patient did not have any serious deep-tissue or lung infections at time of diagnosis at age 3, and it was recognized as an atypical form of CGD -- as no prior patients with NCF4 deficiency and typical CGD infections had mutations involving NCF4. A recently published larger series has borne this out (Van de Geer et al JCI 2018). This study concluded that. "...the patients suffer from hyper-inflammation and peripheral infections, but they do not display any of the invasive. bacterial and fungal infections seen in CGD. Inherited p40phox deficiency underlies a distinctive condition, resembling a mild, atypical form of CGD."

Answer: We thank the reviewer for pointing us to this publication, and now refer to and discuss this study in the revised manuscript, examples:

Introduction (page 3, line 55):

„In a recent review of NCF4 mutations, the clinical presentation of NCF4-deficient patients is described as being even more distinct, resembling a mild, atypical form of CGD⁶.“

Discussion (page 14, line 302):

„...NCF4-deficient patients have normal or mildly impaired PMA-induced neutrophil oxidation^{5,6}.“

Other comments:

7. Abstract and throughout the manuscript:

The study will be better served by describing the results more precisely --- for example, in the Abstract: “.. drastic reduction in the NADPH’s main subunit, gp91phox“ -- is not entirely true – the single PMN sample with Western blot done shows substantial expression of gp91phox, where as in two patients, no gp91phox was detected in monocyte derived macs. – e.g. stating “impaired expression” may be better. (Also, should be “NADPH oxidase’s main subunit” not NADPH’s).

Answer: Throughout the revised manuscript we have attempted to describe our results more precisely. Furthermore, we have specifically addressed the two examples brought up by the reviewer in this comment, by changing „drastic reduction“ to „reduced expression“ and „NADPH’s main subunit“ to „NADPH oxidase’s main subunit“.

8. I disagree with the authors’ statement... “.. that both phenotypic variability and variable expressivity are accepted for the other known CGD genes.” Distinctive features are in fact recognized as a related but different condition. E.g. see above re NCF4 genetic subgroup.

Answer: We acknowledge the reviewer’s view on this matter and have made modifications throughout the manuscript accordingly.

9. The author’s statement in the Introduction, “ The presentation of CGD is highly variable, ranging from relatively mild infections in adulthood to fatal septicemia in infancy”... is not an accurate assessment of the literature.

Answer: We have removed this sentence from the revised manuscript. This paragraph in the Introduction now contains the following text (page 3, line 53):

„Patients with X-linked recessive mutations in CYBB are generally recognized to have the most severe disease course with earlier age at onset, whereas patients with autosomal recessive mutations in NCF1 show a significantly higher age at onset^{1,2,7}. In a recent review of NCF4 mutations, the clinical presentation of NCF4-deficient patients is described as being even more distinct, resembling a mild, atypical form of CGD⁶.“

10.a. In the CYBC1 series presented in this manuscript, the clinical phenotype of the majority of patients does not resemble classic CGD from the infectious disease standpoint – e.g the nature and severity of infection, distinctive spectrum of microbes. The CYBC1 patients are described as having “an excess of atypical and severe infections” -- perhaps these are atypical for a healthy person, but most of these are not characteristic of CGD.

Answer: In the revised manuscript we describe the clinical phenotype of the p.Tyr2Ter homozygous individuals in more detail, pointing out the distinct microbe spectrum and lower frequency of infections. We now devote a paragraph to this in the Results section (page 8, line 170):

*„The overall infectious profile of the homozygous individuals can be considered somewhat unusual for CGD, in terms of the microbe spectrum. The invasive bacterial sepsis by *C.albicans*, and the miliary tuberculosis by *M.tuberculosis* are characteristic of CGD, as well as the pneumonia episodes reported for individual A (at 14 years of age, registered as Legionnaire’s disease, and at 16 years of age). However, the pathogens cultured from the pneumonia episodes are only sporadically reported in CGD (*Legionella*^{1,25} and *S.pneumoniae*⁷), and the absence of some of the most frequently encountered pathogens in CGD, such as *Staphylococcus aureus* and *Aspergillus* species, is noteworthy^{2,7}.“*

Since the last review we received information on a seperate pneumonia episode in individual A, as is stated in the above text, this has also been added to Table 1 of the revised manuscript.

10.b. The index pair of brothers (apart from their significant GI disease,) had some infectious manifestations seen in “classic“ CGD: documented candida septicemia, and episode of lymphadenitis and subcutaneous tissue infections, although no pneumonia or other deep tissue infections. The Burkolderia was cultured from a mouth ulcer, notably, in CGD, Burklderia infections in CGD are typically are invasive lung and soft tissue infections. Legionnaire’s disease , described in one brother, is only rarely described in CGD, so may be a coincidence.

Answer: We agree with the reviewer that both the GI disease and some of their infections could be considered typical for CGD, especially the candidal septicaemia and lymphadenitis. The Burkholderia cepacia was identified in an active wound infection in the mouth, even though it did not result in deep tissue infection. We have modified the following sentence in the manuscript to make this clear (page 4, line 73):

„Both brothers experienced recurrent bacterial infections, including infections around the nose and an active wound infection in the mouth, from which the opportunistic bacterium Burkholderia cepacia was cultured. B.cepacia is known to cause infections in immunocompromised hosts, in particular CGD patients⁴.“

Taking the reviewer's comment into account, we have further modified our discussion of the clinical course of the brothers, noting that CGD patients would typically be expected to have more frequent infections (page 4, line 76):

„At this stage, the combined clinical, histological, and bacteriological evidence led to a suspicion of CGD. A formal CGD diagnosis was subsequently confirmed for both brothers based on PMA-induced neutrophil oxidative burst tests at the time (Fig. 1b). The frequency of infections reported for the brothers was nonetheless somewhat less than what would be expected in X-linked CGD patients.“

Finally, although we agree with the reviewer that Legionella is not the most frequently encountered pathogen in CGD (and now state this in the manuscript, see our above answer to comment 10.a from this reviewer), we feel that the combination of Legionnaires' disease and candidal septicaemia, along with a separate pneumonia episode, cannot be a coincidence for this individual. In the revised manuscript we now address this, and have also added a number showing the rarity of Legionella infections in Iceland (page 7, line 161):

„The rarity of most of these infections is notable, only 61 occurrences of Legionella infections have been reported in Iceland between 1997 and 2017, and only 19 individuals have been diagnosed with candidal septicaemia over the same period²¹. Thus, it is noteworthy that individual A developed both Legionnaires' disease and candidal septicaemia.“

10.c. As far as other patients:

- **One had miliary TB, a pathogen that CGD patients have increased susceptibility to.**
- **Another patient had invasive S. pneumonia as manifested by a positive blood culture. However, S. pneumonia, while described in a few patients with CGD, is not an organism that CGD patients have increased susceptibility to. And, as a group, patients with pneumococcal sepsis are highly unlikely to have CGD.**
- **Increased viral infections are not associated with CGD.**
- **Otitis media, chronic acne or repeated onychomycosis are not deep tissue/ invasive infections.**

Answer: We point to our answer to comment 10.a from this reviewer. Since the last review, we also received information on a separate pneumonia episode in individual A, although the pathogen is unknown. This has now been added to the manuscript (Table 1, and page 8, line 173):

„...pneumonia episodes reported for individual A (at 14 years of age, registered as Legionnaire’s disease, and at 16 years of age).“

In Table 1 we report all confirmed infections for the eight homozygotes. We do not emphasize the viral infections (they are not discussed in the text) but we do note them in Table 1, since we do see viral infections in four out of the eight homozygous individuals.

11.a. That numerous cases with null alleles of CYBB, CYBA, NCF1 and NCF2 have been missed due to an ascertainment bias seems exceedingly unlikely. Conversely, if homozygous CYBC1 could manifest as a classic CGD phenotype, it seems surprising that none in these series had deep tissue or lung infections due to Staph, Aspergillus, or B. cepacia.

Answer: To be clear, when we refer to an ascertainment bias we are talking about missing individuals that carry CGD genotypes and have mild manifestations.

The genotypic set we are working with (155K chip-genotyped Icelanders) consists of individuals over 18 years old at the time of submitting a sample, and these individuals are healthy enough to have survived up to the time of inclusion in the set. Our set is therefore biased towards healthy adults. We have added this clarification to our Discussion section (page 14, line 290):

„The set of 155K chip-genotyped Icelanders used for our study consists of individuals healthy enough to have survived up to the time of recruitment, with the vast majority over 18 years old at the time of sample acquisition (mean age at recruitment is 46 years). The set is therefore biased towards adults, and the identification of genotypes from such a set can be expected to reveal greater phenotypic variability than recruitment based on clinical presentation alone, as is illustrated by our study.“

Our only means of collecting samples from children or affected individuals is through clinical sequencing, as was the case for the two brothers here. It is possible that we have missed children deficient in CYBC1. We do observe three Icelandic couples heterozygous for p.Tyr2Ter in CYBC1 that had offspring that died at an early age. And although we can only speculate that these children were homozygous carriers of p.Tyr2Ter we do note a striking phenotype for at least one of these children, with broncopneumonia reported as a cause of death. To our knowledge there has not been a prior study starting from null genotypes in CYBB, CYBA, NCF1, NCF2, or NCF4. We do however note that siblings of CGD patients who also carry a pathogenic genotype are often asymptomatic until a later age, but are diagnosed early due to awareness of their genotype (and this indeed seems to be the case in the recent review article on NCF4 patients).

We now also discuss the absence of the most frequently encountered pathogens in CGD, i.e. Staphylococcus aureus and Aspergillus species (page 8, line 176):

„...the absence of some of the most frequently encountered pathogens in CGD, such as Staphylococcus aureus and Aspergillus species, is noteworthy^{2,6}.“

11.b. A more likely explanation for the discovery of patients with a genetic defect involving the NADPH oxidase through “recruitment based on genotypes” is that loss of CYBC results in a related but different phenotype because there is residual NADPH oxidase activity, particularly in PMNs, that is protective from infections with microbes characteristic of CGD.

Answer: We refer to our answer to comment 11.a from the reviewer. We would also like to note that the residual ROS seen in the 3 tested CYBC1 p.Tyr2Ter homozygotes is minimal. Whether it is sufficient to provide some protection from characteristic CGD microbes we cannot say.

12.a. This cellular phenotype (which unfortunately is based on relatively limited available data) is consistent with studies in the EROS KO mouse, which was identified originally by its susceptibility to a macrophage pathogen, Salmonella, where intact macrophage function is particularly important. The only other organism the EROS mice were challenged with was Listeria, and again, macrophages play an important role in control. It’s a bit misleading to state that the Eros KO mice show a phenotype similar to ,e.g., CYBB KO mice, with a “high susceptibility to infections”. Only two organisms were studied, and not those where some oxidase activity in PMNs appears crucial e.g. Aspergillus.

Answer: We have now modified the sentence stating that bc017643 knockout show a high susceptibility to infections. The text now reads as follows (page 6, line 116):

„Both neutrophils and BM-derived macrophages from bc017643-knockout mice showed a highly impaired oxidative burst in response to a range of stimuli, including PMA. Accordingly, bc017643-knockout mice (-/-) showed high susceptibility to bacterial infections, evidenced by Listeria monocytogenes and Salmonella enterica serovar Typhimurium infections¹⁴. Interestingly, while the ROS deficit observed in macrophages in the knockout mice was similar to that found in gp91^{phox}/Cybb-deficient mice, some ROS generation was noted in neutrophils from the bc017643-knockout mice but not in the gp91^{phox}-deficient mice¹⁴.“

We also address this point in the Discussion section (page 12, line 257):

„The complete absence of gp91^{phox} from macrophages is also in line with some of the infections observed in the p.Tyr2Ter homozygous individuals, like M.tuberculosis³⁶, Legionella³⁷, and C.albicans³⁸, as well as the increased susceptibility of bc017643-knockout mice to S.Typhimurium and L.monocytogenes infections¹⁴. CYBC1 deficiency thus appears to predispose to a distinct spectrum of microbes that can be considered more specific to macrophages.“

12.b. Interestingly, one patient in the current series had miliary TB,; macrophages are very important for control of Mtb, and infections restricted to mycobacteria were described in several kindreds with point mutations in CYBB resulting in profoundly

impaired gp91phox/ p22phox expression in macrophages but not neutrophils (Bustamante et al Nature Immunol 2011).

Answer: We have now incorporated this interesting aspect into our Discussion section (page 12, line 261):

„CYBC1 deficiency thus appears to predispose to a distinct spectrum of microbes that can be considered more specific to macrophages. Supportive of this, a prior study of two particular CYBB mutations causing infections restricted to mycobacteria demonstrated an impaired oxidative burst in macrophages but not in neutrophils³⁹.“

13. Better acknowledgement in the results/ discussions of the limitations of the cellular phenotype characterization in the current study - very limited analysis of oxidase activity both in number of patients studied and assays performed – only neutrophil DHR assays in two kindreds and to a non-physiologic stimulus, PMA (currently only in the methods). The latter is important as in the EROS mouse, PMN oxidase activity in response to the fungal particle zymosan was only ≈50% reduced. Western blot data on gp91phox expression is also very limited, as previously noted.

Answer: We tested 3 out of 8 homozygous individuals, and all three had severely impaired ROS production, with stimulation indices below 3. We have now added this to the Discussion section of the revised manuscript (page 14, line 299):

„The CYBC1-deficient individuals have strongly impaired PMA-induced neutrophil oxidative burst (stimulation indices lower than 3, as in X-linked CGD⁴², see Fig 1b and Fig 2b)...“

We have further added a section acknowledging the limitations of our analysis (moved and adjusted from the Methods section to the Discussion, page 14, line 303):

„We recognize that for full understanding of the molecular mechanism of CYBC1 deficiency, a more detailed analysis of NADPH oxidase activity is required, such as assessment of particle-induced ROS production, as well as experiments with other cell types.“

14. p. 9 not accurate to say that Individual D is “positive for CGD”...first of all, CGD is a clinical phenotype.

Answer: We have modified this sentence in the revised manuscript (page 10, line 226):

„Individual D is therefore positive for the test widely used as a diagnostic test for CGD without having, to our knowledge, developed colitis or chronic infections at age 30 (see Table 1 and Methods).“

15. p. 10 -only autosomal recessive NCF1 tends to be milder (but with infections involving organisms and sites just like the NCF2, CYBB, and CYBA groups, in contrast to most of the CYBC1 patients); AR NCF2 and CYBA resemble XR CYBB.

Answer: *In the revised manuscript we have modified the text according to the reviewer's suggestion (page 13, line 280):*

„Our results indicate that CYBC1-deficient individuals have some residual ROS production, similar to what has been reported for CGD patients with NCF1 biallelic mutations⁴¹.“

16. Supplement p. 1. Fig 1b is not an intracellular killing assay

Answer: *We thank the reviewer for pointing this out to us, and have removed all references to „intracellular killing“ from the supplementary text.*

Reviewer #2 (Remarks to the Author):

The revised manuscript is much improved and I am satisfied that my comments have been adequately addressed.

I note that the supplementary figures still refer to EROS rather than CYBC1.

Answer: We thank the reviewer for his time and have changed „EROS“ to „CYBC1“ in all supplementary figures.

Reviewer #4 (Remarks to the Author):

This study describes a novel homozygous truncating mutation in the CYBC1 gene in 8 patients with clinical features consistent with a CGD phenotype. The p.Tyr2Ter variant is relatively common in the Icelandic population due to a founder effect. The first two affected siblings were recruited for the study based on their severe clinical phenotype, while the other 6 patients were identified based on their imputed genotype. Their clinical presentation is more variable and ranges from the asymptomatic patient D to two patients with a biopsy proven granulomatous colitis and two patients with recurrent infections not typical for CGD. Detailed clinical records were not available for all patients due to the retrospective data collection. No other candidate pathogenic variants were identified in the index family. None of these 8 patients carry pathogenic variants in other 5 CGD-associated genes or PID-linked genes. The genetic evidence that CYBC1 is a novel gene for CGD is fairly strong. Previous mouse model studies and this study suggest that CYBC1 functions as a chaperone for the cytochrome b-245 complex.

Functional data are somewhat limited due to the lack of primary patients' cells. Experiments were not consistently performed in all cell lineages. Neutrophil oxidative burst defect was confirmed in 3/8 patients, including the asymptomatic patient D. The RNA and protein expression of CYBC1 were consistent with the loss of function mutation identified in these patients. The Authors concluded that CYBC1 may function as a macrophage-specific chaperone for the formation of the NADPH oxidase complex.

This finding will be of great interest to the research community and clinicians. Ultimately, it may help identify other patients with CGD in countries that do not have abilities to do sophisticated population genetic studies.

Answer: We thank the reviewer for constructive comments, and have incorporated some of his comments in our revised manuscript.

Reviewer #1 (Remarks to the Author):

I appreciate the effort taken by the authors to respond to my prior concerns and make additional changes in the manuscript. My concerns have been all adequately addressed in this revision. While more work need to be done to sort out the details of how absence of this chaperone impacts CYBB expression and NADPH oxidase activity in different cell types, this is an interesting and important study that adds to our knowledge of gene defects that can impact the CYBB-containing NADPH oxidase and result in both defects in host defense and inflammatory disorders.

one minor comment - line 171 should be .."invasive fungal sepsis" not bacterial

Reviewers' comments:

Reviewer #1 (Remarks to the Author):

I appreciate the effort taken by the authors to respond to my prior concerns and make additional changes in the manuscript. My concerns have been all adequately addressed in this revision. While more work need to be done to sort out the details of how absence of this chaperone impacts CYBB expression and NAPDH oxidase activity in different cell types, this is an interesting and important study that adds to our knowledge of gene defects that can impact the CYBB-containing NADPH oxidase and result in both defects in host defense and inflammatory disorders.

Answer: We thank the reviewer for the positive feedback.

one minor comment - line 171 should be .."invasive fungal sepsis" not bacterial

Answer: We have modified this sentence in the revised manuscript, according to the reviewer's comment.